# Simultaneous motor preparation and execution in a last-moment reach correction task

K. Cora Ames [1,2], Stephen I. Ryu[3,4,5] & Krishna V. Shenoy[1,4,5,6,7,8]

Motor preparation typically precedes movement and is thought to determine properties of upcoming movements. However, preparation has mostly been studied in point-to-point delayed reaching tasks. Here, we ask whether preparation is engaged during mid-reach modifications. Monkeys reach to targets that occasionally jump locations prior to movement onset, requiring a mid-reach correction. In motor cortex and dorsal premotor cortex, we find that the neural activity that signals when to reach predicts monkeys' jump responses on a trial-by-trial basis. We further identify neural patterns that signal where to reach, either during motor preparation or during motor execution. After a target jump, neural activity responds in both preparatory and movement-related dimensions, even though error in pre-paratory dimensions can be small at that time. This suggests that the same preparatory process used in delayed reaching is also involved in reach correction. Furthermore, it indicates that motor preparation and execution can be performed simultaneously.

[1] Neurosciences Program, School of Medicine, Stanford University, Stanford, CA 94305, USA. [2] Department of Neuroscience, Columbia University Medical Center, New York, NY 10032, USA. [3] Department of Neurosurgery, Palo Alto Medical Foundation, Palo Alto, CA 94301, USA. [4] Department of Electrical Engineering, Stanford University, Stanford, CA 94305, USA. [5] Wu Tsai Neuroscience Institute, Stanford University, Stanford, CA 94305, USA. [6] Department of Bioengineering, Stanford University, Stanford, CA 94305, USA. [7] Department of Neurobiology, School of Medicine, Stanford University, Stanford, CA 94305, USA. [8] Howard Hughes Medical Institute at Stanford University, Stanford University, Stanford, CA 94305, USA. Correspondence and requests for materials should be addressed to K.C.A. (email: kca2120@columbia.edu)

Motor preparation has typically been studied using delayed reaching tasks, in which subjects are told what reach to make before they are asked to move. During the delay between target appearance and go cue, neural activity in motor cortex (M1) and dorsal premotor cortex (PMd) changes in a manner specific to the upcoming movement[1–5]. This preparatory activity is thought to set up subsequent movement-related patterns of neural activity: the preparatory state at the time of the go cue correlates with reaction time (RT)[6,7], and interruption of delay-period neural activity delays RT[8,9], suggesting that time needs to be taken to re-prepare in that case.

Recent work has suggested a further key property of preparatory neural activity: it lies within a different set of dimensions than movement-epoch activity[10,11]. Neurons in motor cortex can seem complicated, often responding in different ways during the delay period and the movement period. However, this complex activity at the level of single neurons can be well-described as a linear combination (or weighted average) of a smaller number of time-varying activity profiles, called neural dimensions (Fig. 1a). In the motor cortex, three primary categories of dimensions have been found, each of which has a distinct computational role. First, preparatory dimensions are occupied during the delay period and have different activity for different reach directions. Activity in preparatory dimensions is thought to serve as the initial condition for generating a particular reach[12–15]. Second, trigger dimensions turn on shortly before the onset of movement[16]. These dimensions precede movement onset in a direction-independent manner: they correlate strongly with when to reach, but do not indicate where to reach. Trigger dimensions are thus thought to transition the system from preparing a movement to generating that movement. Finally, movement dimensions are active during movement and are thought to generate the output patterns required to drive movements[14,17,18] (although not all movement dimensions are necessarily output from the cortex[19]). Thus, the typical delayed reaching steps of prepare-then-reach can be divided into separate computations performed in separate neural dimensions.

Because these dimensions have primarily been identified and studied during delayed reaching, their role in a wider variety of behaviors remains unclear. For example, reaches can be modified online in response to new information, such as a change in target location (target jump)[20–22]. During this online updating process, the concept of "prepare, then trigger, then move," breaks down. Prior recording studies of primates performing this task have found that after a target jump, neurons tend to move from patterns of activity associated with non-jump reaches to the first target to activity associated with non-jump reaches to the final target[20,22–24]. This suggests that the neural process of generating movement is not dramatically different between online corrected reaches and point-to-point reaches.

However, it remains unknown whether these responses are re-prepared or not, as individual neurons often have mixed responses to both preparation and movement[10,11,13]. Furthermore, defining what re-preparation means in the context of online reach correction is itself a challenge. Based on the finding that preparatory and movement activity occur in distinct neural subspaces[10], we operationally define motor preparation to be neural activity in dimensions that are normally active during the delay period. If, following a target jump, movements are re-prepared in the same way as during normal reach preparation, we expect to see a re-entry into these dimensions. If, on the other hand, the new movement is either not prepared or is re-prepared in a fundamentally different manner, we expect to see no special activity in these dimensions.

We use non-jump conditions to identify separate preparatory and movement dimensions in our neural data. We then examine the activity in preparatory and movement dimensions to compare two possible strategies for modifying neural activity after a jump. We term the first hypothesis the Direct Response Hypothesis: neural activity should move directly from a pattern which is appropriate for generating a reach to the first target to a pattern which is appropriate for generating a reach to the final target. If the target jumps before the first reach is triggered, the response should occur in preparatory dimensions, as these are the dimensions which are active at that time (Fig. 1b). In contrast, if the target jumps just before the initiation of movement, the neural response should occur primarily in movement dimensions (Fig. 1d).

We term the second hypothesis the Always Prepare Hypothesis. This hypothesis posits that activity in preparatory dimensions is critical for initializing neural activity in movement dimensions. Thus, the neural corrective process should engage preparatory dimensions regardless of when the target jump occurs. We therefore expect to see a response in the preparatory dimensions not only if the target jumps while preparatory activity is still engaged (Fig. 1c). but also if the target jumps just before the initiation of movement, despite the fact that preparatory activity is not typically engaged at that time (Fig. 1e).

Our findings are consistent with the Always Prepare Hypothesis. We observe a response in the preparatory space after a target jump regardless of when the jump occurs. This suggests that, to change an ongoing reach, the target jump response re-engages the same preparatory process as was used for preparing the original reach. Furthermore, it indicates that movement preparation and movement generation can be performed simultaneously.

## Results

**Behavior.** We trained two monkeys (S, K) to perform a target jump variant of a delayed reaching task (Fig. 2). The monkeys touched and held a center target projected onto a vertical screen. After 500–700 ms, a final target appeared, indicating where they would need to reach. The monkeys were required to withhold from reaching for an additional 0–450 ms (S) or 0–900 ms (K) until the center target disappeared, providing a go cue. On 80% of trials (non-jump trials), the monkeys then reached to the cued target for a juice reward. On the remaining 20% of trials, the target jumped to a new location at a random time after the go cue but before the hand began moving. The monkeys needed to reach to this final target location to receive a reward. Jump trials and non-jump trials were randomly interleaved.

As in previous studies[25–27], reaching behavior depended on how much time passed between the target jump and movement onset (Fig. 3). We examined this transition by fitting a sigmoid to the initial angles of the reaches. If the target jumped right before the monkeys began reaching, the monkeys almost always started reaching toward the first target and needed to correct their reaches online (Fig. 3a, red traces; Fig. 3b). The median time at which monkeys started altering their behavior in response to the target jump was 106 ms (87 ms) for Monkey S (K), as measured by the time the sigmoidal fits crossed 5% of the way from the first to the final angle. As more time passed between the target jump and the beginning of movement, a larger percentage of reaches were initiated toward the final target (Fig. 3a, blue traces; Fig. 3b. For non-normalized angles, see Supplementary Fig. 2). Perhaps surprisingly, the size of the target jump had very little effect on the timing of the behavioral transition between reaching more toward the first target and more toward the final target. Across the jump angles we studied, the time of transition (50% crossing of the sigmoidal fits) was not significantly different (Fig. 3c, d) (one-way ANOVA, Monkey S: $p = 0.86$, $n = 10, 16, 22, 35$ conditions per angle; Monkey K: $p = 0.71$, $n = 6, 7, 3$ conditions per angle), indicating that transitioning a reach between nearby targets was not faster on average than transitioning to far away targets.

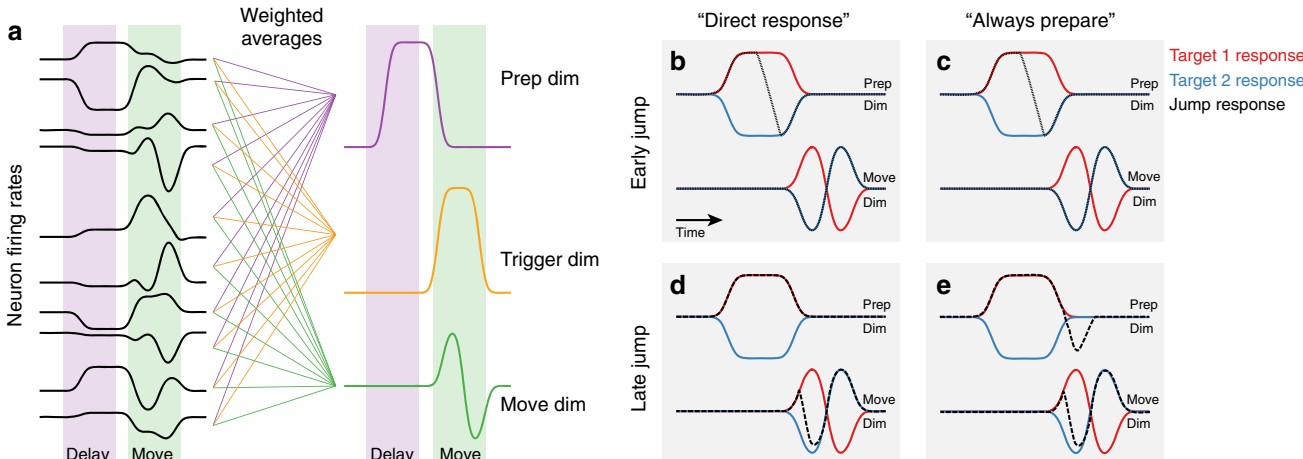

**Fig. 1** Cartoon of neural hypotheses. **a** To identify important signals in the neural population, we can project them into new dimensions. Activity in each dimension is calculated as a weighted average of the firing rates of all neurons. In motor cortex, there tend to be separate dimensions active during movement preparation and movement generation (which tend to have different firing rates across different reaching directions), as well as a trigger dimension which changes in a consistent manner prior to movement for all reach directions. **b** Direct Response Hypothesis, early jumps: For jumps which occur while the motor cortex is still in the preparatory period (before or just after the go cue), neural activity after a jump (dotted black) should transition from preparing a reach to the first target (red) to preparing a reach to the final target (blue). Because the neural correction to the target jump is completed before movement onset, target-jump activity in movement dimensions should be similar to a reach to the final target. **c** Always Prepare Hypothesis, early jumps: For jumps which occur while the motor cortex is still in the preparatory period (before or just after the go cue), the neural correction is in the preparatory dimensions, so the neural predictions of the direct response hypothesis and the always prepare hypothesis are the same. **d** Direct Response Hypothesis, late jumps: For target jumps which occur close to the onset of movement, there is no difference in the preparatory space between the pattern of activity for the first (red) and second jumps (blue). Under the Direct Response Hypothesis, we would therefore expect to see the response to the target jump occur exclusively in the movement-related neural dimensions. **e** Always Prepare Hypothesis, late jumps: Under the Always Prepare Hypothesis, motor preparation must be re-engaged following a target jump. We would thus expect to see a target jump response in the preparatory dimensions, even though these dimensions are not ordinarily active during movement

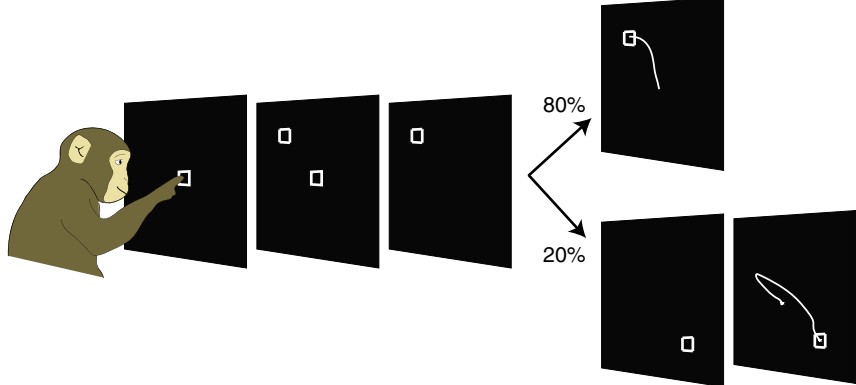

**Fig. 2** Target Jump Task. To initiate a trial, monkeys touched an illuminated center hold target projected on a vertical screen. After 500–700 ms, a final target appeared, indicating where the monkeys should reach next. After a delay period of 0–500 ms (S) or 0–900 ms (K), the center target disappeared, serving as a go cue. On 80% of trials, the monkeys would then reach to the cued target. On 20% of trials (jump trials), the first cued target changed locations at a random time after the go cue but before the monkeys began reaching. The monkeys needed to touch the final target to receive a juice reward

The behavior in target jump conditions can serve as a readout for how quickly motor preparation can be completed; if the monkey has fully re-prepared (or otherwise transitioned to an acceptable set of neural activity to drive a reach to the final target), then he will reach toward the final target. If not, then the reach will begin to the first target. The time needed to correctly re-prepare a reach was substantially shorter than the monkeys' typical RT. The average RT on non-jump trials was 335 ± 49 ms (282 ± 49 ms) for Monkey S (K), whereas the average time at which monkeys finished transitioning to reaching toward the final target following a target jump was 164 ms (160 ms) for Monkey S (K), as measured by the time the sigmoidal fits

crossed 95%. If we directly calculate the probability of reaching to the final target as a function of time from the target jump (instead of using sigmoidal fits to behavior), our results are similar (Supplementary Fig. 3). This suggests that movement initiation time is longer than needed to re-prepare a movement, in agreement with previous studies[25–28].

**Identifying the neural onset of movement.** We recorded neural activity using either 16-channel U-probes in PMd (Monkey S) or two 96-channel Utah arrays, one in M1 and one in PMd (Monkey K). We first examined individual neurons' peri-stimulus

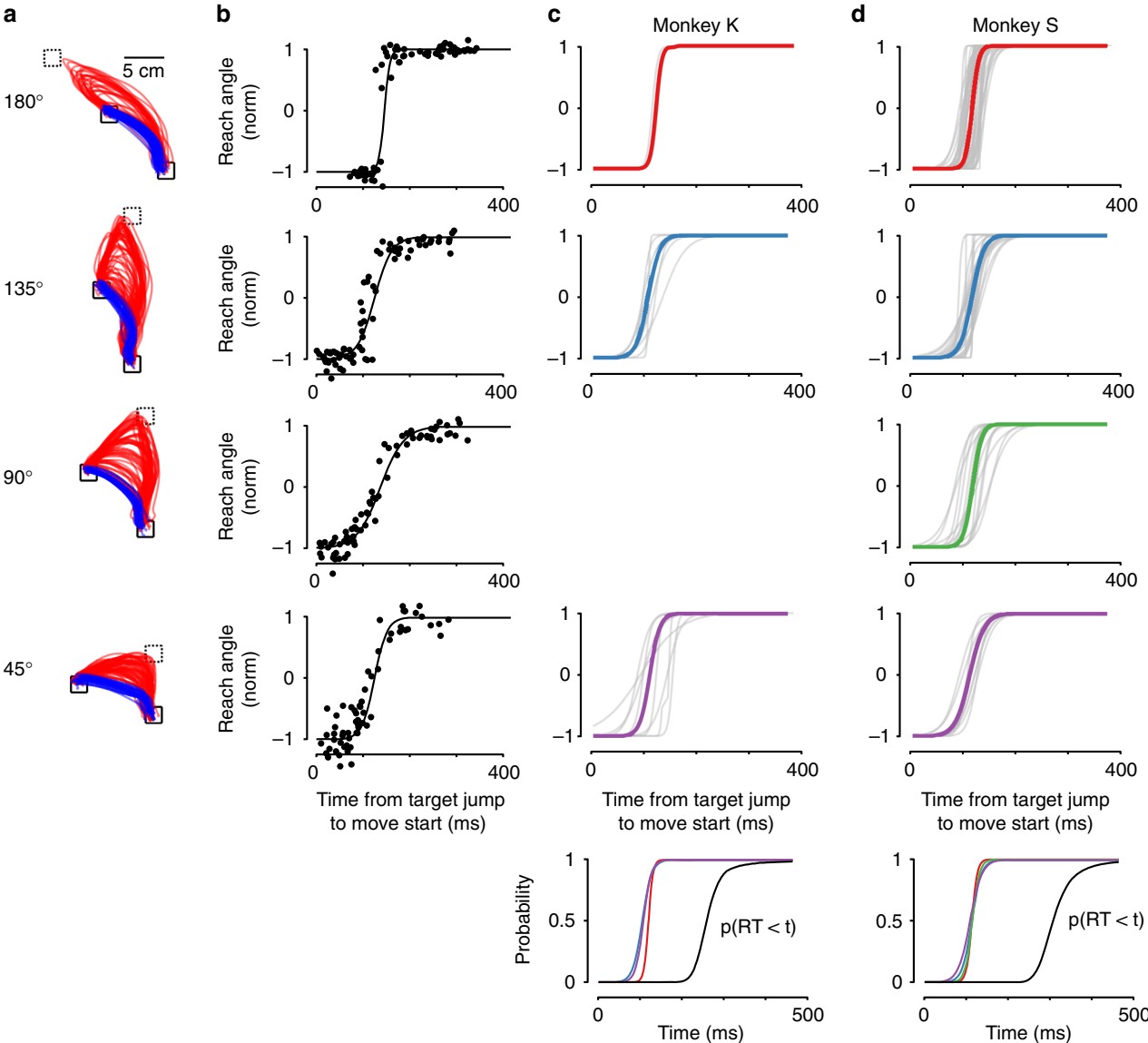

**Fig. 3** Initial reach angles after a target jump. **a** Reach paths after a target jump, for example conditions with a 180 degree, 135 degree, 90 degree, and 45 degree distance between targets. Red traces were initiated toward the first target location and corrected online, blue traces were initiated toward the final target location. **b** Initial reach angle as a function of time from the target jump to movement onset, for the example conditions shown in (a). Each dot shows one trial, lines show sigmoidal fit. **c**, **d** Sigmoidal fits for initial reach angles vs. time from target jump to movement onset, for all recorded conditions, for Monkey K (**c**) and Monkey S (**d**). Colored lines show average fits. Bottom row shows the overlap of average fits for each jump angle (scaled to go from 0 to 1), along with the cumulative RT distribution across all non-jump trials on all recording days. Note that Monkey K did not perform 90-degree target jump conditions, so that entry is left blank. For non-normalized angles, see Supplementary Fig. 2

time histograms (PSTHs). Many units had robust responses to the target jump (Fig. 4). Sometimes, the target jump elicited a simple change of activity, with the unit either increasing or decreasing its firing rate as appropriate (Fig. 4a–c). However, the target jump could also elicit less easily-explained responses. For example, we show a condition where a unit fires more after a target jump than it does when reaching to the first or the final targets in isolation (Fig. 4d). This is true both for reaches initiated toward the first target and for reaches initiated toward the final target, suggesting that the response isn't purely due to different muscle activations needed for on-line reach correction.

Can behavioral performance in target jump trials be predicted based on whether or not the monkey committed to making the first reach before the target jump? Perhaps if the target jump is early enough, there is enough time to change the plan before the

process of generating movement begins. To address this hypothesis, we needed to measure the timing of the commitment to move on a trial-by-trial basis. We leveraged two key results of recent papers. First, neural activity that relates to the transition between movement preparation and movement generation is largely orthogonal to neural activity that distinguishes between different reach directions[10,16]. Second, the dimensions that best predict the timing of movement onset are independent of reach direction[16]. This means that whether or not the monkey is initiating a reach can be predicted the same way regardless of reach direction. This means that we can determine how close the monkey is to moving in the same way regardless of whether the monkey is reaching to the first target or to the final target.

To isolate the dimension which best predicts the time of movement onset, we trained a decoder using all non-jump trials

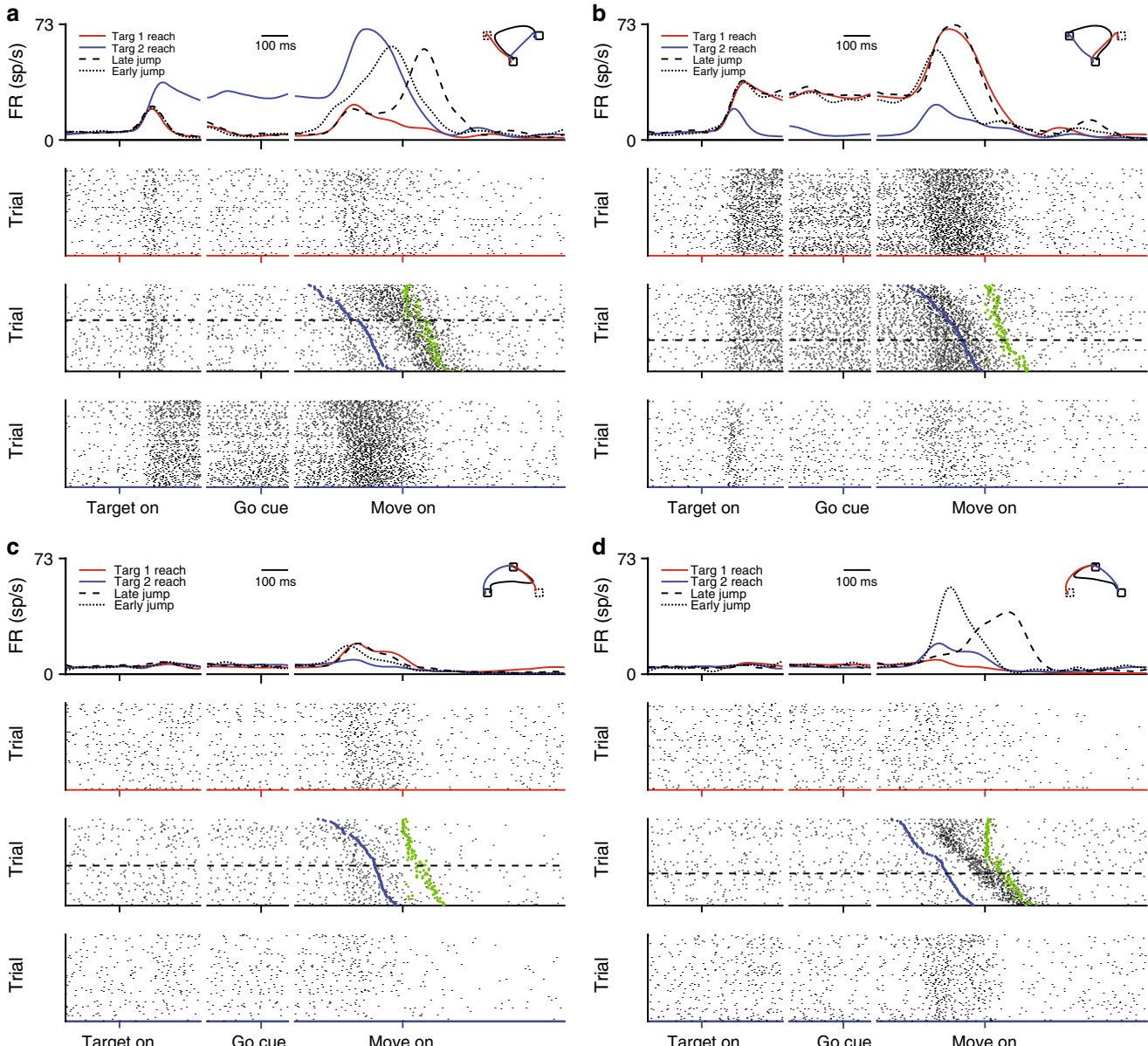

**Fig. 4** Example single unit activity during jump and non-jump conditions. All examples are different jump conditions for the same unit. **a–d** Different example jump conditions, with activity of jump trials shown together with non-jump conditions to the same targets. Top row: Average FR during non-jump reaches to the first target (red), non-jump reaches to the final target (blue), jump reaches which were initiated toward the final target (dotted line), and jump reaches which were initiated toward the first target (dashed line). Second row: Raster plot of spike times during non-jump reaches to the first target. Dimensions are times x trials. Third row: Raster plot of spike times during jumps from the first to the final target. Raster includes both trials in which the hand started toward the final target (above dashed line) and in which the hand started toward the first target and was corrected online (below dashed line). Blue dot indicates the time of target jump. Green dot indicates the time of first detected movement toward the final target. Fourth row: Raster plot of spike times during non-jump reaches to the final target

(combined across all reach directions). We used a support vector machine (SVM) to find the dimension that best distinguishes between neural activity before movement and neural activity around movement onset. Because we require simultaneous recordings for this analysis, we found a separate decoder for each recording day. We excluded datasets with <10 simultaneously recorded units; this removed 0/3 of Monkey K's datasets and 5/24 of Monkey S's datasets.

The decoder performs well at finding the onset of movement in non-jump trials: an average of 86% (99%) of time points were classified correctly on held-out non jump trials for Monkey S (K). We refer to the dimension found by this decoder as the trigger

dimension. On non-jump trials, neural activity begins changing in the trigger dimension around 150 ms prior to the onset of the movement (Fig. 5a, b).

We next examined whether the trigger dimension predicts behavior in target jump trials. For each trial, we calculated the neural trigger event as the first time that neural activity in the trigger dimension crossed zero after the go cue. Note that the location of zero is decoder-dependent, and reflects when the decoder decides that the trial is close to initiating movement. We compared this time to the time of the target jump, to see on each trial whether the neural trigger event occurred before or after the target jump. We found that if the neural trigger happened

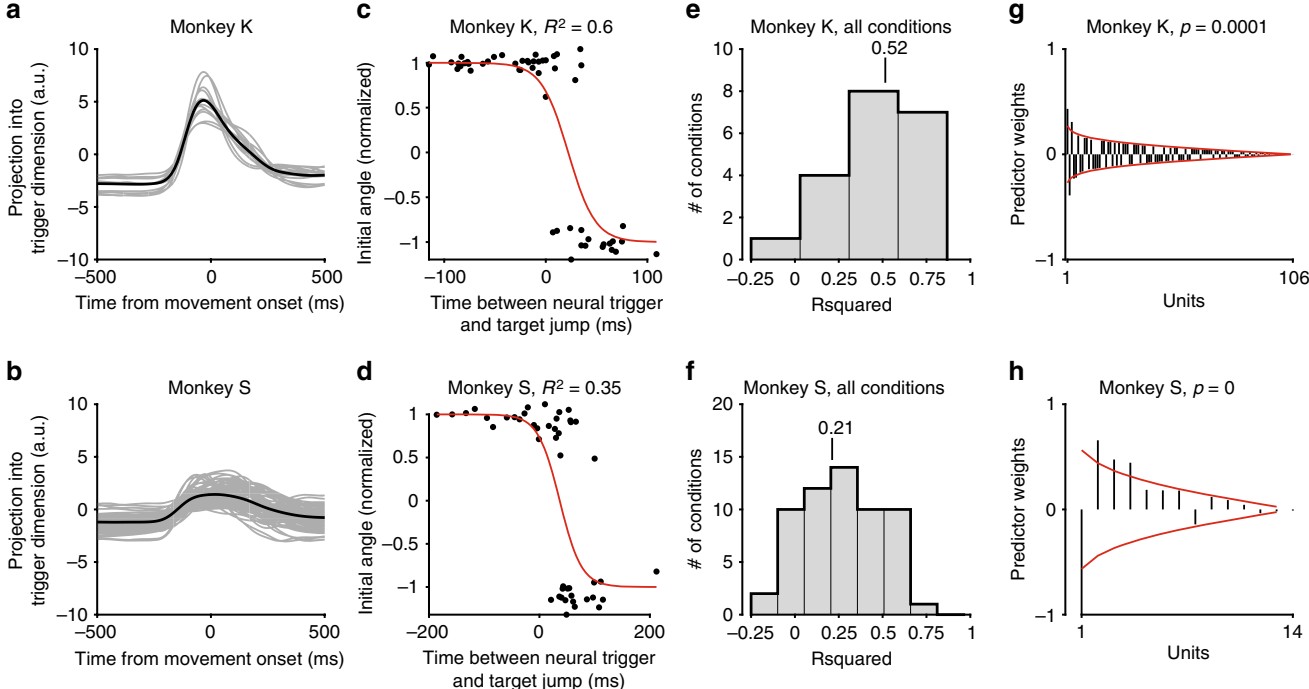

**Fig. 5** Trigger state at time of jump predicts subsequent behavior. **a**, **b** Average neural activity in the trigger dimension, during non-jump trials. Each individual condition is shown in gray, the average across conditions is shown in black. Shortly before movement onset, neural activity begins to change in this dimension. **c**, **d** Example conditions for Monkey K and S. Each dot shows, on the x-axis, the time between the target jump and the time that the trigger signal crosses zero: negative numbers imply that the jump preceded the neural trigger event, positive numbers indicate that the jump occurred after the neural trigger event. On the y-axis is shown the initial reach angle, normalized to range from negative one (first target) to one (final target). Best-fit sigmoidal line is shown in red. **e**, **f** Distributions of quality of sigmoid fit between the difference between neural trigger time and target jump time versus the distance reached to the wrong target, for each jump condition recorded for Monkey K and S. $R^2$ values show generalization accuracy across $n = 20$ conditions (Monkey K) and $n = 68$ conditions (Monkey S). Line shows median accuracy. **g**, **h** Distribution of weights for each unit onto the trigger dimension, for an example dataset for each monkey. Black lines: the magnitude of contribution of each unit to the trigger dimension, ordered by absolute value. Red lines, average weight distributions for random projections

before the target jump, the monkeys were more likely to initiate their reaches toward the first target. If the neural trigger happened after the target jump, the monkeys were more likely to initiate their reaches toward the final target (Fig. 5c, d, example conditions). Across all jump conditions, the relative timing between the neural trigger and the target jump was able to predict a median of 52% (21%) of the variance in the initial reach direction for Monkey K (S), assessed using leave-one-out cross-validation of a sigmoid fit (Fig. 5e, f). Using the same fits, we also analyzed classification accuracy, to determine how well our fits could predict whether a reach would be toward the first or final target. This is a slightly easier problem, as the classifier is simply trying to determine which target the initial reach angle is closer to. We predicted reach behavior with a median of 87% (74%) accuracy for Monkey K (S).

Note that Monkey S's single-trial predictability was worse than Monkey K's (though it could be quite high on some conditions). This is likely due to the fact that each Monkey S dataset included fewer simultaneously recorded units. If we sub-select Monkey K's recordings to match the number of units recorded for Monkey S' datasets, we get similar performance (Supplementary Fig. 4).

We further asked how the weights were distributed across units. We compared the distribution of magnitudes of our projection weights to the magnitudes of weights of 10,000 random projection vectors in the same space (Fig. 5g, h). We found that the kurtosis of the trigger dimension projection (a measurement of the dispersion of the values) differs from random in many of the datasets ($p < 0.05$ in 12/19 datasets for Monkey S and 3/3 datasets for Monkey K), suggesting that our analysis often

relied more heavily on a subset of units for assessing the timing than it would by random chance. We also assessed the distribution of the weights corresponding to units from M1 and PMd in Monkey K (we only had simultaneous M1 and PMd recordings from this monkey). We found that both M1 and PMd units contributed similarly to the trigger dimension, although the weights were somewhat higher on average for M1 than PMd units (Supplementary Fig. 5).

**Target jump responses in preparatory and movement subspaces.** Our behavioral and trigger signal analyses indicate a crucial property of the neural response to a target jump: it is relatively quick, as it can lead to corrected reaching behavior within about 150 ms following the jump. However, it remains to be seen how this correction is accomplished. We therefore examined the neural activity corresponding to where to reach.

A recent report demonstrated that, during delayed reaches, the neural dimensions which are active during preparation are largely orthogonal to the neural dimensions which are active during movement[10]. This provides a useful window onto neural activity, allowing us to separate putatively preparatory activity from putatively movement-driving activity. We leveraged the same method to find orthogonal preparatory and movement dimensions in our data, using our non-jump conditions. We further constrained our preparatory and movement dimensions to be orthogonal to the trigger dimension we found in the previous section. The trigger dimension is concerned with when to reach, whereas here we are interested in isolating the signals related to where to reach. We again found a different set of dimensions for

each recording day, excluding the five of Monkey S's 24 datasets which contained <10 recorded units.

Non-jump reaching conditions can be well-separated into preparatory and movement-related dimensions. During the delay period, activity corresponding to each reach direction spreads out in the preparatory dimensions (Fig. 6a, e), while remaining compact in the movement dimensions (Fig. 6b, f). Around movement onset, the activity for each reach direction spreads out in the movement dimensions (Fig. 6b, f) and contracts in the preparatory dimensions (Fig. 6a, e). To summarize performance, we can look at the cross-condition variance across all preparatory and movement dimensions as a function of time. This variance is higher in the preparatory dimensions during the delay period, and higher in the movement dimensions during the movement epoch (Fig. 6c, g). If we normalize the cross-condition variance by the total cross-condition variance across all dimensions at each timepoint, this tendency is preserved (Fig. 6d, h). During the delay period, preparatory dimensions contain an average of 72% (68%) of the cross-condition variance for Monkey K (S), whereas movement dimensions contain only 1% (11%). During the movement epoch, this effect is reversed: preparatory dimensions contain only 1% (9%) of the cross-condition variance for Monkey K (S), whereas movement dimensions contain 71% (62%). While units from both M1 and PMd contributed to movement and preparatory dimensions, the preparatory-dimension weights were somewhat higher on average for PMd units than M1 units, and the movement-dimension weights were somewhat higher for M1 units than PMd units (Supplementary Fig. 5B, C).

Following a target jump, neural activity needs to change from driving a reach to the first target to driving a reach to the final target. What does this process look like? We first examined how target-jump responses behave in an example preparatory and movement dimension (Fig. 7a, b, e, f). For each jump condition, we calculated each neuron's trial-averaged firing rate for jump trials initiated toward the first target (Late Jumps, black dashed line) and for jump trials initiated toward the final target (Early Jumps, black dotted line). We then projected that activity into the preparatory and movement dimensions. During the delay period, target-jump condition neural activity resembles that of reaches to the first target in the preparatory dimension (Fig. 7a, e). This is expected, as the target jump has not yet occurred. During the delay, activity in the example movement dimension is not strongly active regardless of condition (Fig. 7b, f). After the target jump, neural activity in the preparatory dimension diverges from the pattern for the first target and converges to the pattern for the final target. This convergence is not necessarily direct, especially for jump trials initiated toward the first target and corrected online. For example, in the dimension shown for Monkey K we see an overshoot (Fig. 7a, black dashed line). For Monkey S we even observe an initial response in the opposite direction than expected, increasing rather than decreasing the distance between the firing rates for jump conditions initiated toward the first target (Fig. 7e, black dashed line). In the movement dimensions, in contrast, neural activity seems to move more directly from the non-jump trajectory for the first target to the non-jump trajectory for the final target (Fig. 7b–f).

What is the target jump response across all preparatory and movement dimensions? We can look at the neural distance between the target-jump trajectories and the non-jump trajectories in these spaces. A low value indicates that the target jump activity is similar to that of the non-jump condition, whereas a high value indicates that it is quite different. We examine the distance to the trajectory for non-jump reaches to the final target. Here, target-jump condition distance is initially high, as the monkey is preparing a different reach during the delay period (Fig. 7c, g). After the target jump, neural distance in the

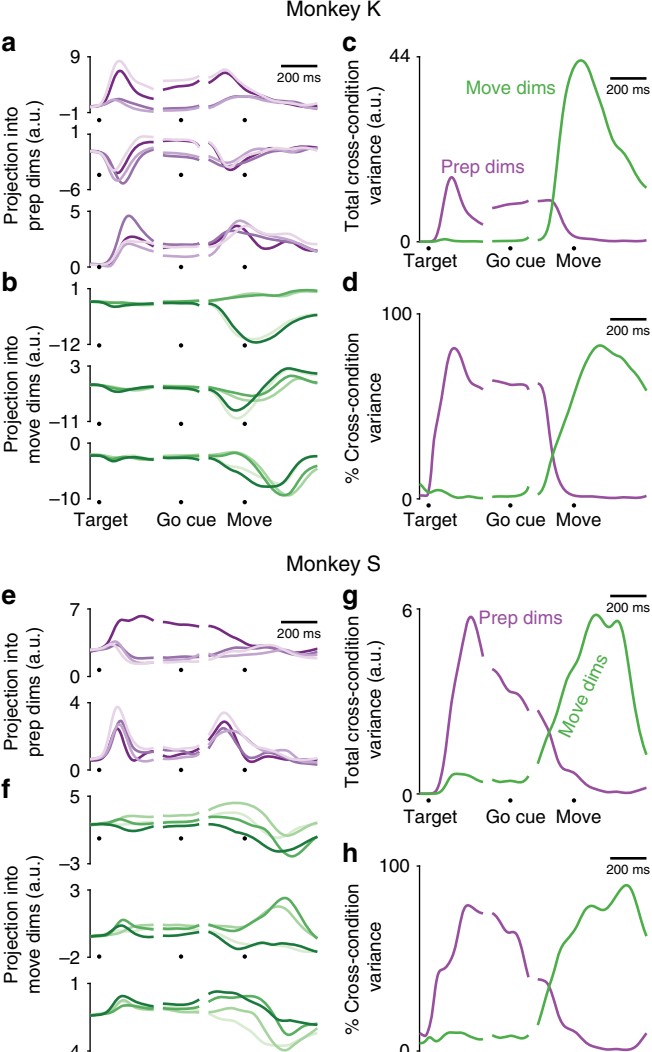

**Fig. 6** Separation of non-jump neural activity into preparatory and movement dimensions. **a, b** For an example dataset from Monkey K, projection of trial-averaged, non-jump reaching condition firing rates into **a** preparatory dimensions and **b** movement dimensions. Each trace is a different reach direction. **c** For the same dataset shown in A-B, the total cross-condition variance across all preparatory dimensions and movement dimensions, as a function of time. Purple: variance in preparatory dimensions. Green: variance in movement dimensions. **d** As in (**c**), but normalized by the total cross-condition variance at each time point. **e–h** As in (**a–d**), for Monkey S

preparatory dimension first increases, then falls away. For reaches initiated toward the first target, the preparatory distance is much larger after the jump than it is between the two non-jump reach trajectories, indicating that the jump response in the preparatory dimensions is disproportionate to the distance between the first and final target trajectories. In the movement dimensions, neural distance increases slightly for both jump behaviors in the time between the go cue and movement; for jump trials initiated toward the final target, the modest distance increase has largely diminished by the time of movement onset, and distance is low during movement (Fig. 7d, h). This is consistent with these jump trials following a similar neural trajectory to that of non-jump trials. The jump trials initiated toward the first target, in contrast, continue to have a high distance at the time of movement onset, as the initial part of the reach is different. In movement

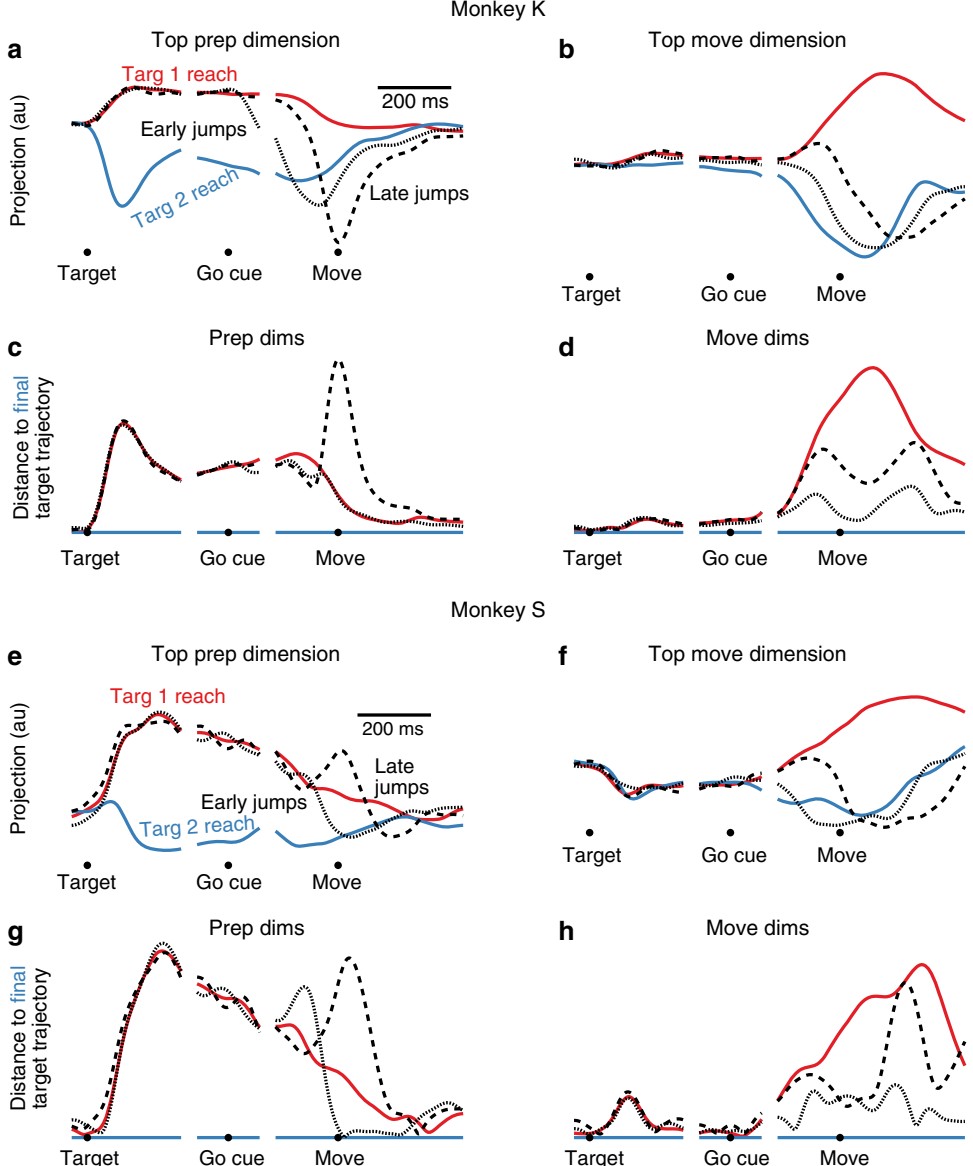

**Fig. 7** Target jump response in preparatory and movement dimensions, example conditions. **a** For an example target jump condition for Monkey K, neural activity in one preparatory dimension. All traces show trial-averaged activity. Red: non-jump reaches to the first target. Blue: non-jump reaches to the final target. Dashed black: Jump trials initiated toward the first target. Dotted Black: Jump trials initiated toward the final target. **b** As in (**a**), but for an example movement dimension. **c** For Monkey K, the time-aligned neural distance to the non-jump reach trajectory to the final target, calculated across all preparatory dimensions. Note that the distance between a trajectory and itself is zero, so the blue line (Distance from a target 2 reach to itself) shows a zero distance. **d** As in (**c**), for distance in movement dimensions. **e–h** As in (**a–d**), for Monkey S

dimensions, the distance falls slowly, and often fails to fall completely to zero; because the overall reach is different between jump trials initiated toward the first target and non-jump reaches to the final target, we do not necessarily expect that the neural distance will fall completely to zero in movement dimensions.

The effects we observed in our example jump conditions remain consistent when we look at activity across all jump conditions (Fig. 8). In the movement dimensions, neural distance increases only modestly between jump trajectories initiated toward the final target and non-jump trajectories for reaches to the final target. Movement-dimension distance remains higher for jumps initiated toward the first target, but nevertheless stays equal to or less than the distance between the non-jump trajectories (Fig. 8c, d). In the preparatory dimensions, jump conditions initiated toward the final target tended not to see a rise in neural distance after a target jump. This is likely because the preparatory-space distance

between jump conditions and non-jump conditions is still high when the target jumps early (Fig. 8a, b). However, for reaches initiated toward the first target, there is a large peak in neural distance in the preparatory dimensions which persists into the movement period. This distance is significantly larger than the distance between non-jump trajectories during movement (Wilcoxon rank-sum across conditions: Monkey K: $p = 6.8 \times 10^{-8}$; Monkey S: $p = 4.6 \times 10^{-5}$). When we separate our conditions by the angle of the reach, the same effect can be seen for all angles except Monkey S's 45-degree jumps (Supplementary Fig. 6).

Furthermore, for late target jumps, note that at the time of movement onset, neural distance is not only high in the preparatory space, but also in the movement space (Fig. 8c, d, dashed line). This indicates that not only are preparatory dimensions re-engaged following a target jump, but also that preparatory dimensions and movement dimensions can be

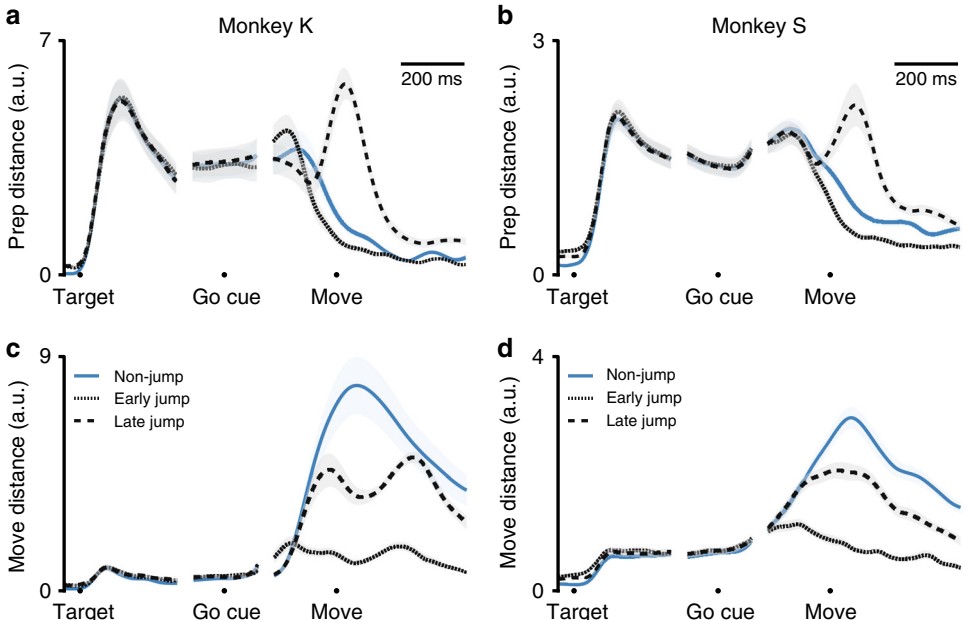

**Fig. 8** Target jump response in preparatory and movement dimensions, across datasets. **a** For Monkey K, neural distance to the neural trajectory for reaches to the final target in the preparatory space. Blue: Distance between neural trajectories for non-jump reaches to the first versus final target. Dotted line: Neural distance for jump reaches which were initiated toward the final target. Dashed line: Neural distance for jump reaches which were initiated toward the first target. All lines are mean ± s.e.m. across conditions. **b** As in (**a**), for Monkey S. **c** For Monkey K, distance to the neural trajectory for non-jump reaches to the final target in the movement space. Colors as in (**a**). **d** As in (**c**), for Monkey S. For conditions separated by reach angle, see Supplementary Fig. 6

engaged at the same time. In other words, the monkey is simultaneously generating movement and preparing movement.

This peak in preparatory distance for late target jumps is inconsistent with the Direct Response Hypothesis. Under that hypothesis, the neural activity should respond to a target jump by changing its activity only along neural dimensions in which activity differs between its first and final target trajectories. If the distance between non-jump trajectories is low in preparatory dimensions, then there is no error to be corrected, and we would not expect to see a jump response in those dimensions. Instead, our results are consistent with the Always Prepare Hypothesis that states that motor cortex must re-engage motor preparation following a last-moment target jump. We see a preparatory response during the movement epoch after a late target jump, even though there is little difference between non-jump reaches in those dimensions at that time.

### Discussion
In this study, we examined neural population signals for where and when to reach during a last-moment target jump task. The neural dimension signaling when to reach (trigger dimension) predicted monkeys' behavior on a trial-by-trial basis. If the monkeys began the neural process of initiating movement prior to the target jump, then they would start their reach toward the first target. If, however, the target jumped prior to neural reach initiation, the monkeys instead began reaching toward the final target.

To study how neural activity transitions from driving a reach toward the first target to the final target, we identified two classes of signals related to where to reach: preparatory and movement dimensions. During ordinary delayed reaching, preparatory dimensions are active only during the delay period, while movement dimensions are active only during movement. We compared two alternate hypotheses. The Direct Response

Hypothesis posits that neural activity will always move directly from the pattern driving the first reach to the pattern driving the second reach. Thus, neural activity should only change in preparatory dimensions if the target jump occurs while these dimensions are still active. The Always Prepare Hypothesis posits that neural activity will always respond in preparatory dimensions, regardless of when the target jump occurs.

Our data were consistent with the Always Prepare Hypothesis. Neural activity responded to the target jump by changing in the preparatory dimensions, regardless of whether the target jump occurred early or late. In the case of jump trials initiated toward the first target, in which movement activity was corrected mid-reach, we actually saw an overlap in time between neural activity in the preparatory dimensions and the movement dimensions.

In this study, we defined preparatory dimensions as the dimensions which are selectively active during the delay period. This definition encompasses two assumptions. First, it assumes that neural activity during the delay period serves a preparatory role. The preparatory role of delay period activity has been heavily studied. Having a delay period speeds RT[29–31], and trial-to-trial variations in delay period activity are related to variations in RT[6,32]. Our finding that these dimensions are also recruited following a target jump adds further evidence that their role is important for programming and updating movement commands.

Our second assumption regarding motor preparation is that preparation will always involve the same dimensions. This again seems reasonable given prior studies of these dimensions. For example, the same preparatory events are activated prior to movement not only during delayed reaching, but also under different behavioral paradigms, including a self-timed movement task and a quasi-automatic response task[33]. However, the same preparatory dimensions need not necessarily have been involved after a target jump. For example, if we had found that neural activity did not re-enter these preparatory dimensions following a target jump, it could have indicated either that re-preparation

does not occur or that re-preparation leverages a different set of neural dimensions. Because we did observe a neural response in these putatively-preparatory dimensions, this supports the hypothesis that these dimensions are, in fact, used not only during the delay period but also during last-moment re-preparation of movements and online reach correction.

A major behavioral signature of target jump responses is that the RT to the jump is faster than the normal RT for point-to-point reaching. This has been observed across a wide variety of behavioral paradigms[21,22,25,34–38], and is similar in magnitude to the RT decrease observed in metronome tasks[28,39]. Two major mechanisms have been proposed to explain this RT speedup.

First, subcortical areas may play a special role during online reach correction. This model is supported by several lines of evidence. First, reach corrections are quite fast (muscle responses can be found within 100 ms[35]), and can be performed subconsciously, for small target jumps that occur during the saccade[25,27,40]. Second, initial jump response behavior is the same regardless of subjects' intention, and only later do task instructions affect behavior[36,41]. Finally, a subject with callosal agenesis experienced an RT deficit when reaching to targets in the opposite hemifield during point-to-point reaches, suggesting that reach initiation normally involves cortical communication pathways. However, target jump responses were unaffected, suggesting a subcortical route for online reach correction[42].

A second proposed mechanism is that motor preparation and execution, normally performed in sequence, instead overlap in time during online reach correction. Evidence that motor preparation and execution can be performed simultaneously come from a few sources. First, Haith and colleagues found that preparation time and initiation time were statistically distinct; indeed, subjects couldn't link the two even when it would be beneficial to do so[43]. Errors in initial reach direction occurred when the reach began before preparation was complete, suggesting that in these error trials, preparation and execution overlapped. Second, neural variability in PMd was reduced following a target jump, suggesting that this area, historically associated with motor planning, may be playing a role in speeding responses to target jumps[38]. Third, a study of reaching where accuracy constraints are only taken into account mid-reach also have a low RT[44]. This suggests that reaches can be initiated with incomplete preparation and prepared online under certain tasks. Furthermore, perturbations of the posterior parietal cortex interfere with the fast response to target jumps, suggesting that cortical visuomotor pathways are involved in online reach corrections[37,45,46], as opposed to reach correction being exclusively the provenance of subcortical structures.

Our results directly examined preparatory and execution related signals in motor cortex, and we found that preparatory signals are re-engaged following a last-moment target jump. This provides new evidence that the same motor preparatory process plays a role not only in specifying a reach ahead of time, but also in online reach modifications. Our results therefore support the model in which preparation and execution can be performed simultaneously during online reach correction. We observe no change in the role of motor cortex for online reach correction versus standard point-to-point reaching; instead, we see the same set of signals in motor cortex during both tasks. Why, then, is the RT so much lower for responses to a target jump than for responses to initial target appearance? Our data supports the hypothesis that normal RTs principally measure the time to trigger the movement, whereas the target jump RT instead measures the time to prepare a movement. This preparation time can be quite fast and thus is normally complete prior to triggering movement[43,47].

Our results are not necessarily incompatible with a role for subcortical pathways. One intriguing possibility is that a subcortical relay of visual information to motor cortex is also used during normal reaching. We previously found that target-related information reaches motor cortex approximately 50 ms after appearing on the screen, suggesting that a very fast pathway transmits these earliest signals. In contrast, the motor cortical response to the go cue is much slower, requiring at minimum 100 ms during a standard reaching task[48]. If the decision of when to reach is more cortically-dependent, then this could explain results like the agenesis study described above[42]. For the subject with callosal agenesis, point-to-point reaching RTs could be impaired when reaching to the opposite hemifield due to an increase in time to trigger the reach, instead of an increase in time to prepare the reach.

To our knowledge, this is the first study to provide neural evidence of simultaneous motor preparation and execution. To identify this effect, we leveraged two critical features of motor cortical activity. First, because movement preparation and movement generation are performed sequentially during delayed reaching[10,43,49], we can isolate time periods where one of these computations is dominant: the delay period for motor preparation, and the movement period for movement generation. Second, because preparation and movement occupy orthogonal subspaces[10,11], we can use these different time periods to identify the different neural dimensions which are active during those times. After a target jump, we can then observe that patterns of neural activity which are typically only seen during the delay period become active during movement. If the brain used the same dimensions for preparation and movement, then it would be much more difficult to distinguish one process from the other.

Beyond experimental convenience, however, our results suggest an important computational advantage to leveraging different dimensions for different computations: the brain gains the ability to perform computations serially or in parallel. Increasing numbers of brain regions have been shown to leverage different dimensions for different computations (e.g., prefrontal cortex[50–52]; posterior parietal cortex[53]; motor cortex[10,11,16,17,33]; locust antennal lobe[54]). Separate computations being mixed at the level of individual neurons but separable at the level of the neural population thus appears to be a common feature. Our results suggest that this separation of signals by dimensions may allow brain regions to alter the temporal relationships between computations: movement preparation and generation can be performed one after another or simultaneously. By studying additional brain regions under a variety of tasks, the ability to alter the temporal relationships between computations performed in different dimensions may be a unifying feature of neural processing.

## Methods

**Behavior**. All research was compliant with ethical regulations for animal testing and research. Research protocols were approved by the Stanford Institutional Animal Care and Use Committee. We trained two male rhesus macaques (macaca mulatta) (K, age 10, 12 kg; S, age 6, 10 kg) to perform a variant of a delayed reaching task (Fig. 2). Each monkey sat in a custom primate chair (Crist Instruments, Inc.) and touched targets projected onto a vertical screen approximately 30 cm in front of them. The position of the monkey's hand was monitored optically using a reflective bead taped between the first and second knuckles of the middle and ring fingers (Polaris, Northern Digital Inc.). Each monkey performed two categories of trials: jump trials and non-jump trials. During non-jump trials, the monkeys touched and held a center target to initiate a trial. After 500–700 ms, a final target appeared 10 cm away from the center target. The monkeys were required to withhold from reaching for a delay period of 0–500 ms (S) or 0–900 ms (K). After the delay, the center target disappeared, providing a go cue. The monkeys could then reach to the peripheral target to receive a juice reward. Jump trials proceeded in an identical manner to non-jump trials through the time of the go cue. At a random time following the go cue but before the minimum RT of the monkey, the first peripheral target would turn off and a second peripheral target

would turn on, at a separation of 180, 135, 90 or 45 degrees from the first target. The monkeys needed to reach to the final target to receive their reward. The range of time intervals between the go cue and the target jump was selected on a day-to-day basis to be no longer than the earliest non-jump RT, and to yield some reaches which were initiated toward the first target, and some reaches which were initiated toward the final target.

Throughout this work, we will use the term jump conditions to refer to any given pair of targets, taking into account order of appearance. For example, a jump from a left target to a right target is a separate condition from a jump from a right target to a left target. To ensure a sufficient trial count for each jump condition, we typically allowed only a subset of possible jumps on any given day. For example, one day might include four targets arranged in a square, but restrict to only 180-degree jumps, leading to a total of four jump conditions.

To characterize the initial angle of each reach, we calculated the angle of the hand position when it crossed a 1 cm radius from the start position. We normalized the reach angles for each jump condition so that an angle of negative one represented a reach toward the first target, and an angle of one represented a reach toward the final target, to ease comparisons across different jump angles. For each jump condition on each day, we then plotted the initial reach angles versus the time between the target jump and the onset of movement, and calculated a sigmoidal fit. In Fig. 3, we show example fits and the average fits for each jump angle that we tested. For all statistics based on sigmoidal fits, we used jump conditions which displayed a fit with a small confidence interval for the 50% crossing time (CI on crossing time must be <50 ms). This ruled out 4/20 conditions for Monkey K and 3/86 conditions for Monkey S. Conditions that were ruled out typically had few reaches that were initiated to the first target, reflecting a jump time distribution that was slightly too close to the go cue. To determine when the monkeys started transitioning their behavior after a target jump, we found the median time that the sigmoidal fits crossed a normalized angle of 0.05. To determine when the monkeys finished transitioning their behavior after a target jump, we found the median time that the sigmoidal fits crossed a normalized angle of 0.95. To assess whether the time to transition behavior was different for different reach angles, we used a one-way ANOVA on the 50% crossing times of the fits (Monkey K: $n = 6, 7, 3$ conditions per angle; Monkey S: $n = 10, 16, 22, 35$ conditions per angle). We also calculated a one-way ANOVA on the slope of the sigmoidal fits, to determine if the transition was sharper in some conditions than in others (Monkey K: $n = 6, 7, 3$ conditions per angle; Monkey S: $n = 10, 16, 22, 35$ conditions per angle).

To calculate RTs for non-jump reaches, on each trial we found the time of maximum reach velocity and then traced backward in time to find the first time that trial's reach velocity fell below 5% of its maximum velocity.

**Neural recordings**. We used two techniques to simultaneously record from multiple neurons in motor cortical areas. Monkey K was chronically implanted with two 96-electrode arrays, one in M1 and one in PMd (Utah arrays, Blackrock Microsystems Inc.). Recording locations were selected using surface features of the brain and information from recordings performed prior to array implantation. We used a total of three array datasets from Monkey K, for a total of 20 reaching conditions. Note that units recorded on chronically implanted arrays are not necessarily different on different days. Monkey K's per-dataset unit count ranged from 76 units to 117 units.

Monkey S was implanted with a cylinder above M1 and PMd. 2.5-mm diameter burr holes were drilled in the skull enclosed by this cylinder, leaving the dura intact. We recorded neural activity with a linear array of 16 electrodes (U-probe, Plexon Inc.) which was lowered acutely each day through one of these burr holes (Supplementary Figure 1). Recording locations for Monkey S were verified using a combination of microstimulation (using single electrodes lowered into the burr hole), palpation, and stereotactic coordinates for the arm area of M1 and PMd. We used an offline spike sorter to identify single-unit and multi-unit activity on each of our recorded electrodes for both monkeys (NeuroSort). Only units with good isolation quality across the day were analyzed. We used a total of 24 datasets from Monkey S, for a total of 288 units across datasets (mean number of units per dataset: 12; std number of units per dataset: 3.9).

All analyses were performed on each day's dataset separately for Monkey S and Monkey K, to leverage the simultaneously-recorded nature of our datasets.

To examine the activity of individual units during reaching (both jump and non-jump conditions), we calculated the firing rate of each unit on each trial by convolving its spike train with a 30 ms gaussian filter. Because dimensionality-reduction analyses can be biased toward over-representing high firing rate units, we normalized each unit's firing rate. We concatenated this unit's activity across all trials, and calculated the standard deviation. We then divided that unit's FR (on each trial) by this normalization value. To ensure that we did not artificially inflate very low-activity units, if the standard deviation of a unit's FR was <1 (which would result in an increase in FR after normalization), we instead set the normalization value to 1, such that these units were not changed by normalization.

**Trigger dimension**. To identify the dimension along which neural activity changes prior to a reach, we trained a SVM. An SVM is a tool to find separating dimensions between classes, optimized to find the best separation between the points which are most similar between groups. The SVM was trained for each day's recordings on single-trial data from non-jump reaches to all targets: it was optimized to

distinguish between points which occurred 360–180 ms prior to movement onset in one category, and points which occurred between 120 ms prior to movement onset and 60 ms after movement onset in the other category. Firing rates were calculated as described above and sampled at 10-ms intervals. Performance was assessed on a test set of 10% of non-jump trials, which were selected randomly and left out of the training set.

To assess the usefulness of the trigger dimension in predicting behavior in jump trials, we found, for each trial, the time that neural activity in the trigger dimension first crossed zero following the go cue. We compared this time to the time of the target jump, to see on each trial whether the target jump preceded or followed the neural trigger event. We then used this time offset between neural trigger and target jump to predict the initial angle of the reach, using a sigmoidal fit. Performance was assessed using leave-one-out cross-validation, in which we trained a sigmoid on all trials but one, and then predicted the behavior on that trial, repeating the process for each trial. $R^2$ values were calculated based on these left-out predictions, using the equation:

$$R^2 = 1 - SS_{res}/SS_{tot} \tag{1}$$

where $SS_{res}$ is the sum squared error of the prediction:

$$SS_{res} = \sum \left( \mathbf{y}_{pred} - \mathbf{y} \right)^2 \tag{2}$$

and $SS_{tot}$ is the sum squared error if the prediction were the mean of $\mathbf{y}$:

$$SS_{tot} = \sum \left( \bar{\mathbf{y}} - \mathbf{y} \right)^2 \tag{3}$$

Note that because the predictions are calculated for trials that the classifiers don't see, the prediction can have a negative $R^2$, indicating that the classifier performs worse than guessing the mean.

To assess whether the trigger dimension relied on only a subset of the units, we assessed whether the distribution of weights was more clustered than expected by chance. We calculated the kurtosis of the trigger dimension, a measure of the dispersion of the values. We then calculated a bootstrap measure of significance by comparing this value to the kurtosis of 10,000 random projection vectors within the same space. The kurtosis was said to be significantly higher than chance if the value exceeded 95% of these random projection vectors.

**Preparatory and movement dimensions**. To identify preparatory and movement dimensions within each dataset, we collected the activity of all of the neurons during each non-jump condition, either during the delay period (0–300 ms from the time of target appearance) or during the movement epoch (0–g rates, such that our final data matrices were of size $n \times (c, t)$, where n is the number of neurons, c is the number of non-jump conditions, and t is the number of timepoints used. To ensure that the dimensions we find for preparation and movement are orthogonal to our previously-found trigger dimension, we first projected the data into the null-space of the trigger dimension, leaving us with matrices of size $(n − 1) \times (c, t)$. While not strictly necessary, this computation reflects the fact that we are principally interested in dimensions which differ across the different conditions, whereas the trigger dimension was selected to find patterns of activity which behave similarly across the different conditions.

We then applied the method developed in Elsayed et al.[10], which simultaneously optimizes for two orthogonal subspaces, one of which maximizes the variance explained during the delay period, and one of which maximizes the variance explained during the movement period. In particular, we optimized the following objective function:

$$\left[ \hat{Q}_{prep}, \hat{Q}_{move} \right] = \arg\max_{[Q_{prep}, Q_{move}]} \frac{1}{2} \left( \frac{Tr\left( Q_{prep}^T C_{prep} Q_{prep} \right)}{\sum_{i=1}^{d_{prep}} \sigma_{prep}(i)} + \frac{Tr\left( Q_{move}^T C_{move} Q_{move} \right)}{\sum_{i=1}^{d_{move}} \sigma_{move}(i)} \right)$$
$$\text{subject to } Q_{prep}^T Q_{move} = 0, \, Q_{prep}^T Q_{prep} = I, \, Q_{move}^T Q_{move} = I \tag{4}$$

$C_{prep}$ and $C_{move}$ are covariance matrices of neural activity during the preparatory and movement epochs. $\sigma_{prep}(i)$ is the $i^{th}$ singular value of $C_{prep}$, and $\sigma_{move}(i)$ is the $i^{th}$ singular value of $C_{move}$. $Q_{prep}$ and $Q_{move}$ are the identified bases for the preparatory and movement subspaces. This technique requires that we specify the number of preparatory dimensions ($d_{prep}$) and movement dimensions ($d_{move}$) ahead of time. We chose the number of dimensions on each day separately, by using principal components analysis to find the number of dimensions required to explain over 70% of the variance during the delay period (which we then used as the number of dimensions for the preparatory space), and over 70% of the variance during movement (which we then used as the number of dimensions for the movement space).

We next assessed the neural response to a target jump in the preparatory and movement dimensions. For each jump condition, we calculated the average firing rates of jump trials initiated to the first target and to the final target. We then projected that activity into the preparatory and movement subspaces, found as described above. We compared the neural activity in target jump conditions to neural activity during non-jump reaches to the first and final target. We calculated the euclidean distance at each time between the trajectories in each subspace at each time point, aligned to the target onset, go cue, and movement onset.

We wanted to determine whether the distance between jump trials initiated toward the first target and the neural trajectory for non-jump reaches to the final target was higher than expected in the preparatory space during the movement

epoch. For scale, we compared the average preparatory space distance between the non-jump trajectories to the first and final targets, from 0–200 ms after movement onset. We then calculated the distance from the non-jump trajectory to the final target, and the average target jump trial trajectory (for jump trials initiated toward the first target). We then calculated a Wilcoxon rank-sum test across the jump versus non-jump distance distributions for all target jump pairs for each monkey to assess significance.

**Reporting summary**. Further information on research design is available in the Nature Research Reporting Summary linked to this article.

## Data availability
All relevant data can be made available by the authors on request.

## Code availability
All relevant analysis code can be made available by the authors on request.

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

## Acknowledgements

We thank Mackenzie Risch, John Aguayo, and Michelle Wechsler for expert animal care and surgical assistance. We thank Beverly Davis for administrative support. K.C.A. was supported by: Stanford Graduate Fellowship, National Science Foundation Graduate Research Fellowship, and a Simons Foundation Collaboration on the Global Brain Postdoctoral Research Fellowship. K.V.S. was supported by the following awards: National Institutes of Health (NIH) National Institute of Neurological Disorders and Stroke (NINDS) Transformative Research Award R01NS076460, NIH National Institute of Mental Health Grant (NIMH) Transformative Research Award R01MH09964703, NIH Director's Pioneer Award 8DP1HD075623, Defense Advanced Research Projects Agency (DARPA) Biological Technology Office (BTO) "REPAIR" award N66001-10-C-2010, DARPA BTO "NeuroFAST" award W911NF-14-2-0013, the Simons Foundation Collaboration on the Global Brain awards 325380 and 543045, an Office of Naval Research award W911NF-14-2-0013 and the Howard Hughes Medical Institute.

## Author contributions

The study was designed by K.C.A. and K.V.S. S.I.R. performed the array implant surgery for Monkey K. K.C.A. trained the monkeys, recorded the data, and performed the analyses. K.C.A. and K.V.S. wrote the paper.

## Additional information

**Competing interests:** The authors declare no competing interests.

