## [Peer Review File · Nature Communications]

Reviewers' Comments:

Reviewer #1:

Remarks to the Author:

Review for Nature Communications:

The interaction of where and when to reach in a last-moment reach correction task.

Significance:

This paper studies neural processes underlying visually guided reach corrections in a planar space. The contribution of this paper is pertinent to the understanding of on-line control kinematics for target-jump scenarios which are known to be highly dynamic. The paper departs from the observation that reaction times for simple reaches are longer than reaction times in on-line corrections. This observation has profound mechanistic implications that are elegantly brought forth. The group proposes two alternative neural mechanisms that could show compliance with behavioural data: a 'sensory gating mechanism' and a 'motor control mechanism'.

The first mechanism implies that target jumps engage non-conscious corrective processes leading to faster sensory processing while the second mechanism implies separate calculations from a slow and 'static' to a fast and 'moving' motor control policies. A 'static' policy has to decide both 'when and where' to move while a dynamic policy has to care only on 'where' to reach for a target jump correction.

Ames et al., collected neurons in M1/PMd using two alternative methodologies, 16 channel U-probe and chronic 96 channel Utah arrays, with similar number of units (270-288) and yields (0.75 units per channel in U probe vs 1 unit per channel in Utah arrays). The techniques and type of data are pertinent to the questions asked.

The group observed that motor areas do not respond sooner to a target jump than to the appearance of the initial target as it would be predicted by the first mechanism.

To address the validity of the second mechanism, the investigators calculated the 'neural movement onset' and the 'neural time of convergence' for target jump corrections, both important variables.

Movement onset was obtained using a known SVM decoder trained on data from direct reaches. This method doesn't require identification of a tuning function or preferred direction alignment for separate neurons and is appropriate for the type of data presented. Neural dimension decomposition allowed Ames et al., to infer movement onset correctly in 83-98% of the time. Examining the state of the best dimension (high trigger dimension state) for movement onset inference, the investigators could predict whether the subject would launch correctly to the target after jump, or whether they would launch in the wrong direction (low trigger dimension state). The investigators showed convincingly that the best dimension was rather explained by the entire neuron population rather than by subsets of units, which is a compelling result towards the robustness of data and analysis conducted. To obtain the time of 'neural convergence' the authors calculated the 'minimum Euclidean neural distance' between jump trials and direct reach trials. Ames et al., found that only for some selective dimensions the correction takes place early, as required per behaviour, while for many others the corrections take place late. This is not incompatible with the second motor control mechanism proposed albeit it is more reminiscent of optimal feedback control. This aspect too is brought forth pertinently in the discussion.

The notion that separate neural dimension decomposition can be involved in different aspects of reaching control constitutes the core of this paper. Using separate neural dimensions the authors can extract from the same population information decoding the temporal initiation of a reach and the correction to a target jump. Since different neural dimensions refer to a dynamic pool of linear combinations of neurons, sequential or parallel computations such as required per a 'when' and

'where' specification are flexible and can largely adapt to the task demands.

It was a pleasure to review this well written paper and I recommend it for publication with only a few minor comments and additional questions.

QUESTIONS

Comparing motor areas:

In the paper neurons from motor areas M1 and PMd are mentioned indiscriminately in movement onset and convergence analysis. Is there any informative difference to report between timings for these two areas? How does adding premotor cortex data improve SVM classification? Also in terms of power, which of the two areas contributes better to each pertinent neural dimension classification?

Characterizing the minimum number of units required for SVM classifiers:

Although the authors convincingly demonstrate that particular subsets of neurons do not contribute particularly more than the entire population (Distribution of weights for the Trigger dimension shows a quite homogeneous population) yet it is unclear what is the minimum number of units required to obtain a conducive classification. A diagram showing the accuracy of decoding the correct movement onset (% correct classification) as a function of number of neurons would be instrumental in addressing this question.

GENERAL COMMENTS

Introduction

The statement in line 43-44 needs clarification and sounds dismissive,

43. 'However, these studies have not concentrated on examining the timing of this neural transition relative to the behavioral transition and thus are not sufficient to explain the time difference between reaches...'

These studies do report temporal measures in the form of behavioural reaction times and single neuron latencies, but these measures have limitations, which could be cited more explicitly, considering the information that a neural state requires.

Population assemblies of individually sampled single neurons do not characterize well the neural state of the population composed of simultaneously recorded units in a single-trial basis. The amount of information that individual cell recordings can convey is limited because the variance across the population on a single-trial basis is not known. In contrast, the information collected in a single-trial basis across several simultaneous collected units constitutes a significant improvement for completeness over the previous methods and allows the characterization of the neural state of the network

Rephrase the first portion of 43 to state these studies have partially examined the timing...and add a brief clarification of the limitations as above whilst explaining more clearly the advantages of characterizing the neural state

54. end. 'We next leveraged the results of a recent report...'

Include citation or eliminate sentence

Results Figures and Legends

Figure 4. Legend

Each panel has 4 rows, first row compares 4 conditions :

1st condition non-jump reaches to initial target (red), 2nd non-jump reaches to final target (blue), jump reaches initiates towards final target (dotted line) and jump reaches initiated toward the initial target (dashed line).

The other 3 rows are rasters that compare 3 of the 4 conditions, one of them is unclear: 6th line.

Raster plot ... jumps from the first to the second target. Is this equivalent to the condition in which the jump reaches where initiated towards the initial target? If this is the case replace the wording. Please consider adding the raster plot for the target jump reaches initiated towards the final target for completeness

6th line. Each row is a trial. Third row. The row terminology is confusing. The term row in panel and row in the raster appear next to each other without further explanations. Please clarify row in the raster or eliminate this sentence altogether.

Results Main Text

Figure S3 is not referred anywhere in the main text perhaps it could be mentioned in line 311-312 365. Time of convergence was also determined with the simple latency measures in Pastor-Bernier 2012. (190ms after the GO signal, Figure 3C, Top 2nd column in page 7) which is consistent with the 204ms and 324ms obtained with the minimum Euclidean measures cited in this study. Please incorporate and rephrase accordingly.

Methods

491-492 Adding the 5 to the denominator.

A better explanation for choosing such an arbitrary number is required. The range of a FR could be described by its intrinsic standard deviation which is part of commonly used Z-score normalization across trials: $Z = (FR - \text{Mean Baseline}) / \text{std}(FR)$. If the range of the average rate described in this soft-normalization is equivalent to the range covered by the standard deviation this would be less hardwired. It is likely that soft-normalizing expression as described could be plausible for the dynamic range observed for signals from motor/premotor cortex but it might not generalize to other motor areas with different dynamic ranges such as found in basal ganglia. Please justify the use of this type of soft-normalization and show its pertinence over other more common types of normalization.

References

3. DJ Crammond and JF Kalaska. This Reference appears concatenated 3 times
36, 30 Indicate volume and page for these publications

Reviewer #2:

Remarks to the Author:

The submitted paper by Ames and colleagues examines a the neural basis of a notable phenomenon in the reaching and grasping literature. That is, that participants respond to a target's change in location faster than they respond to its initial appearance. By recording from population of neurons in M1/PMd in two monkeys, they attempt to resolve between two possible explanations for this phenomenon. First, that the neural systems that update target information engage a pathway different from that which generates the initial reach. Second, that because participants are already engaged in the act of reaching - which may the time consuming bit - they are able to save time by merely updating where they are reaching. Overall, I think this is a timely paper that nicely leverages modern analytical techniques to interrogate the neural control of reaching. I have several comments related to the motivation and experimental design that, in my opinion, require the authors to significantly revise some of their claims and rework the presentation of the material.

Major Comments

1. My biggest concern is that the paper is sub-optimally constructed to actually test the hypotheses put forward. This is on two important fronts. First, the hypothesis. If hypothesis one is correct, you would need to record in some other parts of the nervous system that could form the bypass. Given the

motivation of the paper, that might be posterior parietal cortex or perhaps in the brainstem. The authors have not done that, so they cannot fully rule this solution out. Second, the task. If the motivation is really to examine the behaviors previously investigated with respect to target jump (from Preblanc on down which permeate the citation list) then why not train the monkey to do the actual task that these people did? These two issues give the paper a strange framing that sets it up for under-delivering, which it does. That is not to say that the paper is not interesting, it is. My feeling is that the interesting bits, basically related to Figure 7 (and the last few sentences of the Abstract) need to be front and center. This can be done by focusing on their ability to separate the responses to where and when, a technique introduced in their previous Neuron paper. This, of course, tempers the outputs of this paper but I don't think that's much of a problem as the specific advance is still notable and will be easier to appreciate.

2. The reaction time of the animal to jump trials seems quite slow relative to previous papers. This becomes less of an issue if #1 above is taken into account but it speaks to the degree to which this task is the same thing as the other work in this topic.

3. How is it that the motor cortex begins changing its activity to the initial target 43 and 20 ms after visual target onset for the two monkeys. Both seem very fast but the second one is strikingly so. Perhaps the authors can give more details here as looking at the Neuron paper suggests something like 50ms in those two monkeys. In fact, Monkey K seems to be in both of these studies. Please clarify when making claim, consistent with out previous work.

4. Figure 6 is complicated and needs to be better motivated. A,B appear to be forced by their definitions since the authors have effectively searched for neurons that behave this way. The first in C,D seem very weak, especially D. Even then, should these not be forced through the origin - if that is done the fit for S will be non existent. I appreciate that this may be due to less data in that monkey but that is not a great argument in my opinion for such a central claim.

Other Comments

1. I think it is worth noting that rapid updating of reaching movements is not vision specific. A recent paper by Pruszynski and colleagues (Current Biology, 2016) shows that tactile information about reach location can be used to similar effect. This finding may speak to the possible neural mechanisms at play - esp. about the role of visual and posterior parietal cortex (related to Major #1 above).

2. Although the authors mention it to some degree, I think it would be more natural to lead with the Heath et al paper (2015) as a motivating idea. As mentioned above, I am not sure the current hypothesis driven construction is the way to go so a rephrasing around this recent idea may be a more natural way to get into this material.

3. The authors talk about "sinusoidal" fits with regards to Figure 2 (both in the legend and in the main text). I assume this is sigmoidal.

Reviewer #3:

Remarks to the Author:

General Comments:

It has been established that when subjects are asked to change the direction of the target they are researching to, subjects react faster to the target change than to the initial target appearances. This

study examines the neural correlates of this behavioral phenomena in the motor cortices of Rhesus monkeys using a target-jump task. The main findings are that the response time to the appearance of the jump target is not different from the initial target, suggesting the decrease in reaction time is not due to faster sensory processing. Instead, the results show that the cortical activity can be parsed into orthogonal dimensions processing "when to reach" and "where to reach". The activity corrected more quickly in the dimension that best separated the original reaches from new reaches. The authors argue for sequential and parallel processing in the motor cortex dependent on task requirements.

The study will be of interest to motor control and system neuroscientists that are interested in how information is processed in neural assemblies and, more specifically, motor cortical function. The study continues to develop the intriguing dimensional and neural state analyses begun in previous publications. However, there are several general and specific issues that need to be addressed.

The first general comment is that the manuscript does not frame the Introduction or Discussion in terms of the existing or new theories of motor cortical function. What are the implications for the major theories of how the primary motor cortex (MI) or premotor cortex (PM) function and their roles in movement? How does addressing this problem improve our understanding of the role of the motor cortices in motor control? Furthermore, is there any evidence/literature that suggests the increase in the reaction time to a target jump is a function of either MI and/or PM?

The second major comment is the final argument and data analyses presented in relation to Figure 7. The goal is to separate the neural activity into two components, the neural activity that best separates the initial and final targets (C and E) and the remainder of the activity. Concerning Figure 7, it appears that several of the plots are either misplaced or potentially wrong (the reader assumes the former). Plot A is nearly identical to plot F and Plot B is nearly identical to Plot D in both shape and variability. Therefore, it seems that plot D and F are incorrect, that is plots for the two monkeys have been switched. (As stated above the other possibility is that the plots are incorrect, but the reader assumes just switched.) The second issue is that for both monkeys, the plots of the dimension separating the initial from final reach (C and E) is apparently only a small fraction of the overall mean neural distance (A and B). The questions are whether the fraction of the firing in this dimension is actually only a small component of the firing, can this be better quantified, and, if so, how can we be certain of its overall importance to the behavior.

The third major comment concerns the analysis of the initial reach angle and analyses presented in Figure 2B. The question is how was the data normalized and is the normalization affecting the conclusions? It appears that the normalization was based on the degree of the jump (i.e. 180 vs 45 degrees). However, if this is the correct understanding, the normalization appears to be distorting the data. For example, while the plots for 180 degrees show few in-between values, the jumps for 45 degrees appear distributed with more intermediate values. However, normalizing by 180 will make the initial angles small, actually 4 times smaller than normalizing by 45 degrees. For example, a value of 0.1 in the 180 degree plot is equivalent to 0.4 in the 45 degree plot. Finally, the data in Figure 2 was stated to be fit to a "sinusoidal" function. Clearly, the shape of these plots are not sinusoidal. Were the data actually fit to a sigmoid?

The final general comment is that the manuscript will be a challenge for non-specialists. Particularly for Figure 5 and beyond in which the neural state concepts and dimensional analyses are introduced and used. These concepts and analyses rely heavily on the authors' previous publications, which in themselves are challenging. An effort should be made to better explain the neural state and dimensional analysis as this would increase the readability of the manuscript.

Specific Comments:

1. The description of the times of the analyses in lines 515 to 518 is confusing and should be restated.
2. Why is there such a large difference in the number of datasets for the two monkeys? Why did monkey K with the linear arrays only have 3 datasets? Does that mean just three recording days, which seems very limited. A clearer definition of a "dataset" would be helpful.
3. Several of the statistical analysis rely on sequential testing of time points to determine the timing of a change. However, sequential testing has an inherent problem with multiple testing on non-independent data, i.e. if you test enough times you will get a significant change. How was the multiple testing problem controlled for?
4. Both primary motor cortex and dorsal premotor cortex neurons were recorded and analyzed together. Are there any differences in the neural activity in the two areas? Wouldn't one expect PMd neurons to respond earlier and therefore, may respond differently to the target jump.
5. On page 11, line 229 the prediction median was 39% and 15% for the two monkeys. These numbers seem low, particularly so for the second animal. It suggests that the neural state analysis represents only a fraction of the information needed. This needs additional comments.
6. In Figure 6F, there are negative R2 values? How is this possible?
7. Figure 5A shows the data for the argument that the neural response time does not differ between the onset and jump response, therefore, the sensory processing is not different in the two conditions. However, in both monkeys, the target jump response rises to a higher level and remains higher. The accumulation of neural activity over time is greater for the target jump. Isn't it possible that the detection or perception of the target depends on the accumulated response and, therefore, the accumulation

Reviewer #4:

Remarks to the Author:

In this paper, Ames and colleagues investigate the difference in reaction time between trials where the target jumps just after the go cue and trials where it did not. It had been reported that reaction times are lower for jump trials than non-jump trials but the reason for this difference remains unclear. The authors found that the decoupling between the process of where to reach and when to reach was responsible for this effect and that the facilitation effect found in the jump trials was due to the fact that the go-process was already active and need not to be restarted later.

I like the approach and the study brings new insights to an interesting phenomena. Yet, I think that there are some conceptual problems with the study and that the analysis of the data could be better performed.

Major concerns.

1) The authors compared the activity in jump and non-jump trials. Yet, the activity after the first target appearance must be confined to the null-space (Kauffman) in order to prevent any motion from the arm before the go cue. In contrast, after the target jump, the neural activity should not be confined to the null space and can already take place in all dimension given that the movement must not be prevented. Therefore, I wonder whether the neural activity after the presentation of the first target and the neural activity after the jump are really comparable. Indeed, the dimensions that can be explored in these two moments are different (null space vs. the whole space). The ideal control would have been non-jump trials without delays where neural activity can also evolve in the null and movement sub-spaces but these trials are absent from this report.

2) The first model presented by the authors (faster processing along the visuomotor transformation) looks like a strawman model to me. Indeed, if we look at the effect of attention on sensory processing, people do not report faster activity (despite the fact that attention leads to faster reaction time) but to higher firing rate at the single neuron level and more correlated firing rates at the population level (Reynolds, John H. and Leonardo Chelazzi. 2004. "Attentional Modulation of Visual Processing." Annual Review of Neuroscience 27(1):611-47; Maunsell, J. H. R. and E. P. Cook. 2002. "The Role of Attention in Visual Processing." Philosophical Transactions of the Royal Society B: Biological Sciences 357(1424):1063-72). Nobody has ever demonstrated that the faster reaction time due to attention were due to faster transmission time. Therefore, the first model proposed by the authors does not make sense to me. Interestingly, the authors observed one hallmark of attention in their data, an increased firing rate. I wonder whether the correlated activity also increases in their data after target jump as compared to after target presentation. If it were the case, the facilitation of reaction time after target jump would look a lot like an attentional effect at the neural level and this should be reported.

3) The analysis linked to Fig.6c and d seems very indirect to me. The authors want to look at (lines 208-210) "how close the monkey is to making a movement, and this readout should tell us (the same information) whether the monkey is reaching to the initial target or to the final target.". Yet, they do no report correlation between those two variables but between other variables that are somehow related to them. Indeed, rather than looking at the time of the target jump with respect to the neural trigger moment, they look at the state of the trigger signal at the time of target jump. This does not seem to be a good variable as it looks very variable (Fig 6a and b). I don't see any reasons why the time relative to the time of neural trigger cannot be directly measured. The other variable was a measure of the reaching movement in the wrong direction. The authors chose for the travelled distance. Yet, this variable can be highly influenced by the vigor of the movement. Comparing the actual reaching direction to the reaching direction of non-jump trials to the final target is certainly a better measure to quantify how far from the final target the monkey is going towards.

4) Detection of the different timing measure are based on the significance and non-significance of t-tests. This method is completely inappropriate and suffers from the fact than one outlier could make the test non-significant by increasing the SD. The authors should rather use the ROC analysis technique first used by Brian Corneil (Corneil, Olivier and Munoz, Neuron, 2004). This technique is also used by Steve Scott in his papers. In comparisons to what the authors have been using to detect differences between curve, the ROC technique will be much less sensitive to individual outliers and is therefore better suited to detect timing of different events. All the corresponding analyses need to be changed.

5) Analysis related to Fig 7 c-f is really complicated to follow because the criteria for convergence for the different panels are different . In 7c and e, 1 means neural activity close to initial target and -1 means close to final target (but this isn't explained properly in the main text). In contrast for 7d and f

(and also 7a and b), convergence corresponds to going to zero. I feel that this difference on the definition of convergence in the same figure is really confusing. Moreover, the description and meaning of these curves need to be improved (e.g. by adding for 7c and e that 1 corresponds to neural activity for movements towards the initial target and -1 towards final target). Here are several questions related to the above analysis (Fig.7):

1) I wonder whether the faster detection of changes in neural activity in the separating dimension could be due to the higher signal to noise ratio there. Clearly, with the t-tests used here, a better SNR would yield an earlier detection of the convergence. So, is the SNR comparable in the two dimensions?

2) The criteria for detecting convergence is very different in the two dimensions. For the separating dimension, crossing zero is sufficient (even though the convergence is not completed yet). For the other one, the minimum distance should be zero. I believe that the authors could use the same criteria in both cases (minimum distance on each time point could be computed in the separating dimension as it was done in the null dimension). This would yield a fairer comparison for the timing of convergence between the two dimensions.

3) The authors should comment on the relative timing of neural activity convergence and the timing of movement onset. Right now, it is difficult to know whether this convergence happens before or after movement onset as everything is aligned on target jump time.

We would like to thank the reviewers for their insightful suggestions and attention to detail. We have significantly modified the manuscript in response to their suggestions.

In particular, we have transitioned away from a focus on the timing of neural activity, in order to focus more carefully on the results which previously were examined in our figures 7 and 8. Specifically, we focus on the role of three key sets of neural dimensions in this target-jump response task: the trigger dimension (which was included previously), and “preparatory” and “movement execution” related dimensions (which were previously not explicitly separated). We have entirely re-written the introduction and large parts of the discussion and added a conceptual figure to help with explaining our analyses and hypotheses for a broad audience. We have removed several of our previous figures (those with a focus on timing), and added new data figures (6-8) focusing on distinguishing the role of preparatory and movement related dimensions in motor cortex. We have also slightly modified our data presentation for our behavioral figure (now figure 3) and example raster plot figure (now figure 4), and modified our analyses for the trigger dimension (now figure 5), as per reviewers’ suggestions.

We have addressed the reviewers’ suggestions to the best of our ability, and we believe that the manuscript has been substantially improved by the incorporation of these modifications. We include specific responses to each of the reviewers’ comments below.

Color coding:

Black: Reviewers’ comments.

Blue: Our responses.

Green: Excerpts from the paper.

Reviewers' comments:

Reviewer #1 (Remarks to the Author):

Review for Nature Communications:

The interaction of where and when to reach in a last-moment reach correction task.

Significance:

This paper studies neural processes underlying visually guided reach corrections in a planar space. The contribution of this paper is pertinent to the understanding of on-line control kinematics for target-jump scenarios which are known to be highly dynamic. The paper departs from the observation that reaction times for simple reaches are longer than reaction times in on-line corrections. This observation has profound mechanistic implications that are elegantly brought forth. The group proposes two alternative neural mechanisms that could show compliance with behavioural data: a ‘sensory gating mechanism’ and a ‘motor control mechanism’.

The first mechanism implies that target jumps engage non-conscious corrective processes leading to faster sensory processing while the second mechanism implies separate calculations

from a slow and ‘static’ to a fast and ‘moving’ motor control policies. A ‘static’ policy has to decide both ‘when and where’ to move while a dynamic policy has to care only on ‘where’ to reach for a target jump correction.

Ames et al., collected neurons in M1/PMd using two alternative methodologies, 16 channel U-probe and chronic 96 channel Utah arrays, with similar number of units (270-288) and yields (0.75 units per channel in U probe vs 1 unit per channel in Utah arrays). The techniques and type of data are pertinent to the questions asked.

The group observed that motor areas do not respond sooner to a target jump than to the appearance of the initial target as it would be predicted by the first mechanism. To address the validity of the second mechanism, the investigators calculated the ‘neural movement onset’ and the ‘neural time of convergence’ for target jump corrections, both important variables.

Movement onset was obtained using a known SVM decoder trained on data from direct reaches. This method doesn’t require identification of a tuning function or preferred direction alignment for separate neurons and is appropriate for the type of data presented. Neural dimension decomposition allowed Ames et al., to infer movement onset correctly in 83-98% of the time. Examining the state of the best dimension (high trigger dimension state) for movement onset inference, the investigators could predict whether the subject would launch correctly to the target after jump, or whether they would launch in the wrong direction (low trigger dimension state). The investigators showed convincingly that the best dimension was rather explained by the entire neuron population rather than by subsets of units, which is a compelling result towards the robustness of data and analysis conducted. To obtain the time of ‘neural convergence’ the authors calculated the ‘minimum Euclidean neural distance’ between jump trials and direct reach trials. Ames et al., found that only for some selective dimensions the correction takes place early, as required per behaviour, while for many others the corrections take place late. This is not incompatible with the second motor control mechanism proposed albeit it is more reminiscent of optimal feedback control. This aspect too is brought forth pertinently in the discussion.

The notion that separate neural dimension decomposition can be involved in different aspects of reaching control constitutes the core of this paper. Using separate neural dimensions the authors can extract from the same population information decoding the temporal initiation of a reach and the correction to a target jump. Since different neural dimensions refer to a dynamic pool of linear combinations of neurons, sequential or parallel computations such as required per a ‘when’ and ‘where’ specification are flexible and can largely adapt to the task demands.

It was a pleasure to review this well written paper and I recommend it for publication with only a few minor comments and additional questions.

Thank you for this accurate and comprehensive summary, as well as for the kind words.

QUESTIONS

Comparing motor areas:

In the paper neurons from motor areas M1 and PMd are mentioned indiscriminately in movement onset and convergence analysis. Is there any informative difference to report between timings for these two areas? How does adding premotor cortex data improve SVM classification? Also in terms of power, which of the two areas contributes better to each pertinent neural dimension classification?

Thank you for your question. We did not see dramatic differences between M1 and PMd in our analyses, which is why we have grouped the data together for the paper. However, we agree that care must be taken when doing so, and we have therefore added a supplemental figure whose goal is to illustrate the contribution of M1 and PMd to the specific neural dimensions we analyzed in this paper (Supplemental figure S5, also included below for reference). In this figure, we analyze data from Monkey K, where M1 and PMd units were recorded simultaneously. We find that the trigger dimension pulls roughly similarly from M1 and PMd units (with a slight bias toward stronger weights from M1 units, while preparatory dimensions rely more strongly on PMd units and movement related dimensions weight M1 more strongly, on average, consistent with there being stronger delay-period activity in PMd compared with M1, on average.

Figure S5: Distribution of weights across M1 and PMd units. In Monkey K, units were simultaneously recorded in M1 and PMd. We therefore examined the trigger dimension weights for units from Monkey K's datasets to determine if one area preferentially contributes to the trigger, prep, and move dimensions. (A) Each panel shows the distribution of the absolute value of trigger dimension weights across units, for M1 (orange) and PMd (green). Lines indicate the medians of the distributions. P-values calculated using a two-sided Wilcoxon rank sum test. (B) As in A, for the preparatory dimensions. (C) As in A and B, for the movement dimensions.

Characterizing the minimum number of units required for SVM classifiers:

Although the authors convincingly demonstrate that particular subsets of neurons do not contribute particularly more than the entire population (Distribution of weights for the Trigger dimension shows a quite homogeneous population) yet it is unclear what is the minimum number of units required to obtain a conducive classification. A diagram showing the accuracy of decoding the correct movement onset (% correct classification) as a function of number of neurons would be instrumental in addressing this question.

Thank you, this is a very helpful point. We are aided in this analysis by the fact that the "trigger signal" tends to be one of the largest signals present in the motor cortex during reaching (Kaufmann et al., 2016). Still, as you point out, there is likely some minimum number of units required to ensure a reliable isolation of this signal from the population. To address your question, we sub-sampled each of our datasets to determine how well our SVM classifier identified the trigger dimension (accuracy assessed on left-out trials). We found that even with as few as two units, we exceeded chance accuracy. However, accuracy asymptotes close to 100% around 50 units, with accuracy of about 85% with only 10 units. Supplemental figure S4 has been updated to show these analyses, and the figure is included here as well.

Figure S4: Accuracy of trigger dimension identification versus number of units used. To determine whether prediction accuracy differences between monkeys K and S could be caused by fewer simultaneous units in Monkey S, we performed repeated sub-selections of units from Monkey K and repeated our trigger-signal analysis. We first assessed how well neural data from held-out non-jump trials could be classified using a subset of 2,5,10,25, or 90 units (Monkey K), or a subset of 2,5, or 10 units for Monkey S. (A) Average performance for each dataset, across 10 resamplings for each size. Orange lines: Monkey K. Blue lines: Monkey S. (B) Median performance across all datasets and resamples. Shaded areas show 10th and 90th percentiles of the distribution.

GENERAL COMMENTS

Introduction

The statement in line 43-44 needs clarification and sounds dismissive, 43. ‘However, these studies have not concentrated on examining the timing of this neural transition relative to the behavioral transition and thus are not sufficient to explain the time difference between reaches...’

These studies do report temporal measures in the form of behavioural reaction times and single neuron latencies, but these measures have limitations, which could be cited more explicitly, considering the information that a neural state requires.

Population assemblies of individually sampled single neurons do not characterize well the neural state of the population composed of simultaneously recorded units in a single-trial basis. The amount of information that individual cell recordings can convey is limited because the variance across the population on a single-trial basis is not known. In contrast, the information collected in a single-trial basis across several simultaneously collected units constitutes a significant

improvement for completeness over the previous methods and allows the characterization of the neural state of the network

Rephrase the first portion of 43 to state these studies have partially examined the timing...and add a brief clarification of the limitations as above whilst explaining more clearly the advantages of characterizing the neural state

54. end. 'We next leveraged the results of a recent report...'

Include citation or eliminate sentence

Our apologies for inadvertently coming across as dismissive or otherwise not paying full and proper attention to previously published studies. To address this we have significantly revised the introduction, in response to suggestions from yourself and other reviewers as well. Our reports of previous studies in our introduction now reads:

To examine the role of trigger, preparatory, and movement dimensions in online reach modification, we trained monkeys to perform a target jump task. In this task, the monkeys are told to prepare a reach to a target during the delay period, but on a subset of trials this target can jump locations after the go cue but before movement onset. This task has been used in both humans (Soechting and Lacquaniti, 1983; Prablanc and Martin, 1992; Henis and Flash, 2004) and monkeys (Georgopoulos et al., 1981, 1983; Archambault et al., 2009, 2011; Pastor-Bernier et al., 2012; Dickey et al., 2013) to study the process of last-moment reach modification. Subjects can typically respond quickly to the target jump, reaching toward the new target location if the jump precedes movement by more than about 150 ms. If there is less than 150 ms between the target jump and movement onset, then subjects will instead reach toward the first target and correct their reach online. Neural studies of primates performing this task have found that after a target jump, neurons tend to move from patterns of activity associated with non-jump reaches for the first target to activity associated with non-jump reaches for the second target (Georgopoulos et al., 1981, 1983; Archambault et al., 2011; Pastor-Bernier et al., 2012). This convergence is relatively quick; in PMd, divergence from the initial trajectory can be detected beginning by approximately 120-150 ms after the jump (Wise and Mauritz, 1985; Archambault et al., 2011; Pastor-Bernier et al., 2012), and convergence with the new neural activity was observed to take just under 200 ms (Pastor-Bernier et al., 2012). In M1, the responses to the target jump were observed to take slightly longer, with initial responses recorded on average around 150-180 ms post-jump (Georgopoulos et al., 1983; Archambault et al., 2011). While these studies did a thorough job characterizing the responses of individual units to a target jump, individual neurons often have mixed responses to both preparation and movement (Churchland et al., 2010; Kaufman et al., 2014; Elsayed et al., 2016b). Population-level analyses are thus needed to distinguish the role of preparatory and movement signals in this task.

Results Figures and Legends

Figure 4. Legend

Each panel has 4 rows, first row compares 4 conditions :

1st condition non-jump reaches to initial target (red), 2nd non-jump reaches to final target (blue), jump reaches initiates towards final target (dotted line) and jump reaches initiated toward the initial target (dashed line).

The other 3 rows are rasters that compare 3 of the 4 conditions, one of them is unclear: 6th line. Raster plot ... jumps from the first to the second target. Is this equivalent to the condition in which the jump reaches where initiated towards the initial target? If this is the case replace the wording.

Please consider adding the raster plot for the target jump reaches initiated towards the final target for completeness

6th line. Each row is a trial.

Third row. The row terminology is confusing. The term row in panel and row in the raster appear next to each other without further explanations. Please clarify row in the raster or eliminate this sentence altogether.

Thank you for letting us know that our descriptions of this figure were confusing. We have changed the figure and legend in the following ways. When introducing the raster plots, the description now reads "Dimensions are times x trials." This should avoid over-loading the word "row." In the target jump raster, both trials initiated toward the first and toward the second target are included. For trials initiated toward the first target, the time of movement correction is indicated by a green marker. To clarify this, we have included a dividing line between the trials initiated toward the first vs. second target, and this section of the legend now reads: Raster plot of spike times during jumps from the initial to the final target. Raster includes both trials in which the hand started toward the final target (above dashed line) and in which the hand started toward the initial target and was corrected online (below dashed line). Blue dot indicates the time of target jump. Green dot indicates the time of first detected movement toward the final target.

Results Main Text

Figure S3 is not referred anywhere in the main text perhaps it could be mentioned in line 311-312

Thank you, this was an oversight. We have added a reference to Figure S3 at the end of the paragraph you suggested. The sentence reads: Activity in the separating dimension looks similar for all target jump angles, although better separation is achieved for targets which are further away (Figure S3).

365. Time of convergence was also determined with the simple latency measures in Pastor-Bernier 2012. (190ms after the GO signal, Figure 3C, Top 2nd column in page 7) which is consistent with the 204ms and 324ms obtained with the minimum Euclidean measures cited in this study. Please incorporate and rephrase accordingly.

Thank you for pointing out this additional important connection to a prior study. We have attempted to include more details in our introduction regarding connect points with the important prior work on humans and primates in this task, and we explicitly include mention of the neural response timing results from prior studies in this section.

Methods

491-492 Adding the 5 to the denominator.

A better explanation for choosing such an arbitrary number is required. The range of a FR could be described by its intrinsic standard deviation which is part of commonly used Z-score normalization across trials: $Z = (FR - \text{Mean Baseline}) / \text{std}(FR)$. If the range of the average rate described in this soft-normalization is equivalent to the range covered by the standard deviation this would be less hardwired. It is likely that soft-normalizing expression as described could be plausible for the dynamic range observed for signals from motor/premotor cortex but it might not generalize to other motor areas with different dynamic ranges such as found in basal ganglia. Please justify the use of this type of soft-normalization and show its pertinence over other more common types of normalization.

We used this type of soft-normalization, as dimensionality reduction methods such as PCA will often over-represent high-firing rate units. The number 5 was selected as a kind of “cutoff” to ensure that our small fraction of very low-FR units (which tend to be quite noisy) do not have their activity blown up (which could cause the low-FR units to be over-represented, instead).

However, we note that the exact form of normalization is not critical to our findings. Thanks to your suggestion, we have changed to z-score normalizing our firing rates instead, which, as you note, may make the analysis method more easily portable to studies of brain regions with different firing rate profiles.

References

3. DJ Crammond and JF Kalaska. This Reference appears concatenated 3 times
36, 30 Indicate volume and page for these publications

Thank you, we have now made this correction.

Reviewer #2 (Remarks to the Author):

The submitted paper by Ames and colleagues examines a the neural basis of a notable phenomenon in the reaching and grasping literature. That is, that participants respond to a target's change in location faster than they respond to its initial appearance. By recording from population of neurons in M1/PMd in two monkeys, they attempt to resolve between two possible explanations for this phenomenon. First, that the neural systems that update target information engage a pathway different from that which generates the initial reach. Second, that because participants are already engaged in the act of reaching - which may the time consuming bit - they are able to save time by merely updating where they are reaching. Overall, I think this is a timely paper that nicely leverages modern analytical techniques to interrogate the neural control of reaching. I have several comments related to the motivation and experimental design that, in my opinion, require the authors to significantly revise some of their claims and rework the presentation of the material.

Thank you for this accurate and comprehensive summary.

Major Comments

1. My biggest concern is that the paper is sub-optimally constructed to actually test the hypotheses put forward. This is on two important fronts. First, the hypothesis. If hypothesis one is correct, you would need to record in some other parts of the nervous system that could form the bypass. Given the motivation of the paper, that might be posterior parietal cortex or perhaps in the brainstem. The authors have not done that, so they cannot fully rule this solution out. Second, the task. If the motivation is really to examine the behaviors previously investigated with respect to target jump (from Preblanc on down which permeate the citation list) then why not train the monkey to do the actual task that these people did? These two issues give the paper a strange framing that sets it up for under-delivering, which it does. That is not to say that the paper is not interesting, it is. My feeling is that the interesting bits, basically related to Figure 7 (and the last few sentences of the Abstract) need to be front and center. This can be done by focusing on their ability to separate the responses to where and when, a technique introduced in their previous Neuron paper. This, of course, tempers the outputs of this paper but I don't think that's much of a problem as the specific advance is still notable and will be easier to appreciate.

Thank you for this important observation and very helpful guidance. In response to this comment, and the comments of other reviewers, we have significantly revised the introduction to better set-up the results related to separating the role of different neural dimensions in this task, and their contributions to behavior. We have also significantly expanded our treatment of the separation of the roles of different neural dimensions in this task, as opposed to focusing as heavily on the timing of neural responses in our previous version. We hope that you find this to be a more compelling framing of our core results; we think it will given your helpful and specific guidance.

2. The reaction time of the animal to jump trials seems quite slow relative to previous papers. This becomes less of an issue if #1 above is taken into account but it speaks to the degree to which this task is the same thing as the other work in this topic.

Thank you, we agree that it is important to comment on this in the paper. The RT differences between our (previous) Figure 3 and previous papers is largely due to how RT was calculated. In many papers (e.g. Haith et al 2015), RT to the target jump was calculated as the length of time required after the target jump (or target appearance in the case of “metronome tasks”) for most reaches to be initiated in the direction of the new target. Our results agree with previous studies with this measurement, finding the RT to be about 150 ms. In our prior version of Figure 3, we instead looked at RT as being the first time after the target jump that we observed movement toward the new target: either the time of movement onset, for trials initiated correctly, or the time of reach correction, for trials initiated incorrectly. However, this has the effect of making the apparent RT longer.

Because we think that the important variable is the typical time required to initiate a reach in the correct direction, we have eliminated our prior reaction time figure (previously figure 3), and instead included a panel in our behavior figure (current Figure 3) which explicitly compares the probability of correctly initiating a reach after a target jump to the distribution of non-jump reaction times. We also include Supplemental Figure S3, which calculates the probability of a correct response to a target jump calculated directly (rather than from the sigmoidal fits, as in the main paper).

We hope that this will be both less misleading, and bring our behavioral results in better alignment with previous reports.

3. How is it that the motor cortex begins changing its activity to the initial target 43 and 20 ms after visual target onset for the two monkeys. Both seem very fast but the second one is strikingly so. Perhaps the authors can give more details here as looking at the Neuron paper suggests something like 50ms in those two monkeys. In fact, Monkey K seems to be in both of these studies. Please clarify when making claim, consistent with out previous work.

This has to do with how neural FR is calculated: smoothed with a gaussian or with a boxcar filter. We had smoothed with a gaussian to enable more consistent FR comparisons between convergence and divergence, but this means that this analysis is somewhat acausal. Thanks to your and other reviewers suggestions, we have reduced our emphasis on the timing of the neural responses, in order to expand and highlight our treatment of the role of different neural dimensions to the response to a target jump. We therefore no longer include this figure in our paper. However, we would like to note that, when firing rates are calculated with a causal filter (for example a 30-ms boxcar filter), it brings the timing more in line with previous reports.

4. Figure 6 is complicated and needs to be better motivated. A,B appear to be forced by their definitions since the authors have effectively searched for neurons that behave this way. The first in C,D seem very weak, especially D. Even then, should these not be forced through the origin - if that is done the fit for S will be non existent. I appreciate that this may be due to less data in that monkey but that is not a great argument in my opinion for such a central claim.

Thank you for pointing out this area where additional work is needed. We have made some modifications to Figure 6 (now Figure 5) which we believe improve the issues you point out.

First, we made a slight alteration in the way we predicted behavior based on the trigger signal. Rather than looking at the state of the trigger signal at the time of the target jump, we instead identified the trigger signal's zero-crossing time for each trial, and compared that time to the time of the target jump. This more precisely measures the variable in question: did the target jump before or after the monkey began the neural process of initiating the first movement? Second, we predicted the initial angle of the reach rather than the distance reached to the wrong target. While these two values are related, we feel that predicting the initial angle enables a closer correspondence with previous behavior analyses (such as in Figure 1).

We would like to note, however, that the "zero" crossing point of the trigger signal is partially influenced by our training procedure - the support vector machine is trained to separate times (in non-jump trials) that are from 340-180 ms prior to reach onset versus times that are from 100 ms before to 60 ms after reach onset. As a result, the best time for a zero crossing is approximately 140 ms prior to movement onset. If we shifted these windows, it is likely that a similar dimension would be identified, but that the location of "zero" may be altered. We therefore caution against assigning too much meaning to the exact placement of the "zero" value of the trigger signal.

Other Comments

1. I think it is worth noting that rapid updating of reaching movements is not vision specific. A recent paper by Pruszynski and colleagues (Current Biology, 2016) shows that tactile information about reach location can be used to similar effect. This finding may speak to the possible neural mechanisms at play - esp. about the role of visual and posterior parietal cortex (related to Major #1 above).

You raise an interesting and important point about other tasks which may be relevant for investigating the neural process of online reach correction. We have added a paragraph in our discussion touching on the possible relevance of our findings to mechanical perturbations of the arm, which we include below:

In this study, we concentrate on the neural responses to a visual shift in target location. However, there are many forms of perturbations to which subjects may need to respond during movements. For example, subjects also make corrective movements in response to tactile or proprioceptive information. If a change in target location is signaled via tactile information, subjects can respond with similar latency and accuracy as for visual target jumps (Pruszynski et al., 2016). Furthermore, subjects also make corrective movements to perturbations of the limb, typically at a latency even shorter than the latency for corrections in response to visual perturbations of the cursor or target (Cluff et al., 2014). While the shortest-latency responses to mechanical perturbations likely reflect spinal circuitry (Ghez and Shinoda, 1978), long-latency mechanical perturbation responses can be quite sophisticated, taking task requirements into account (Scott, 2012). The Optimal Feedback Control paradigm suggests that a major function of preparatory activity is to set the feedback control policy for the upcoming movement (Scott, 2004; Todorov and Jordan, 2002). We found that movements are re-prepared following a target jump. One possible hypothesis would be that this change in task goal requires a similar change in control policy, therefore requiring re-preparation. A different form of perturbation, such

as a small mechanical perturbation of the arm, may not require a change in policy and thus may not re-engage preparatory activity. Alternately, preparatory may need to be engaged for any change in the ongoing reach. Additional experiments under a wider variety of perturbations may help to shed light on this question.

2. Although the authors mention it to some degree, I think it would be more natural to lead with the Heath et al paper (2015) as a motivating idea. As mentioned above, I am not sure the current hypothesis driven construction is the way to go so a rephrasing around this recent idea may be a more natural way to get into this material.

Thank you for this suggestion. We have substantially changed the focus of our paper, to concentrate more closely on the role of different neural signals related to “where” and “when,” to reach, rather than focusing as heavily on the timing. We have re-written the introduction to lead with a discussion of these signals (split further into the trigger signal for “when” and preparatory/movement signals for “where”). While we did not have space for a thorough discussion of the Haith results in the introduction, we reference them several times in the results and include a paragraph in the discussion considering our connect points with this study. This paragraph reads:

Recent behavioral work has suggested that the neural computations of “where” and “when” to reach are independent of each other. When human subjects are forced to reach at times close to the target appearance, they are able to make accurate reaches beginning about 150 ms after target onset (Ghez et al., 1997; Haith et al., 2015). This latency is much faster than typical reaction times, and of a similar latency to reaction times to target jumps (Goodale et al., 1986; Soechting and Lacquaniti, 1983; Prablanc and Martin, 1992). Haith and colleagues further observed that the distributions of “minimal preparation” times and “response initiation times” were consistent with motor preparation and movement initiation being separate, independent computations. Subjects could speed their reaction times by beginning reach initiation more quickly, if encouraged to do so with strict reaction time penalties. However, this faster reaction time came at the expense of more errors; subjects did not seem to be able to speed motor preparation in a similar way. These results suggest that typical reaction times are longer than they need to be, in order to ensure that the movement is always prepared properly. Our results lend further support to the idea of separate signals corresponding to the decision of “where” and “when” to move. We found that the behavior in this task relates to separate “where” and “when” computations, which are shifted relative to each other in time. For example, during normal reaching, motor preparation always precedes the neural trigger and movement generation. For late target jumps, however, motor preparation can occur after the trigger has been pulled, overlapping in time with movement generation. The relative timing of the target jump with the movement trigger signal suggests that, as long as the jump occurs before the decision to move, the re-preparation process is fast enough to be complete. Furthermore, we were able to find neural signatures of both “where” and “when” to reach in motor cortex, indicating that the signals in this region may be sufficient to reveal the current state of these computations, although these computations are likely to involve other regions, as well.

3. The authors talk about “sinusoidal” fits with regards to Figure 2 (both in the legend and in the main text). I assume this is sigmoidal.

You are correct, we intended to say “sigmoidal.” All references to “sinusoidal” fits now have been corrected to read “sigmoidal.”

Reviewer #3 (Remarks to the Author):

General Comments:

It has been established that when subjects are asked to change the direction of the target they are researching to, subjects react faster to the target change than to the initial target appearances. This study examines the neural correlates of this behavioral phenomena in the motor cortices of Rhesus monkeys using a target-jump task. The main findings are that the response time to the appearance of the jump target is not different from the initial target, suggesting the decrease in reaction time is not due to faster sensory processing. Instead, the results show that the cortical activity can be parsed into orthogonal dimensions processing “when to reach” and “where to reach”. The activity corrected more quickly in the dimension that best separated the original reaches from new reaches. The authors argue for sequential and parallel processing in the motor cortex dependent on task requirements.

The study will be of interest to motor control and system neuroscientists that are interested in how information is processed in neural assemblies and, more specifically, motor cortical function. The study continues to develop the intriguing dimensional and neural state analyses begun in previous publications. However, there are several general and specific issues that need to be addressed.

Thank you for this accurate and comprehensive summary.

The first general comment is that the manuscript does not frame the Introduction or Discussion in terms of the existing or new theories of motor cortical function. What are the implications for the major theories of how the primary motor cortex (MI) or premotor cortex (PM) function and their roles in movement? How does addressing this problem improve our understanding of the role of the motor cortices in motor control? Furthermore, is there any evidence/literature that suggests the increase in the reaction time to a target jump is a function of either MI and/or PM?

In our new manuscript, we have attempted to more closely link to current theories of motor cortical function. In particular, we examine a key hypothesis of the role of preparatory activity in M1/PMd: that it is necessary to have the correct preparatory activity in order to set up the right movement execution (as reviewed in Shenoy et al, 2011). We have moved away from focusing as much on the timing of the behavior in order to provide more room to explore and discuss the behavior of preparatory, movement, and trigger signals in this task. We also consider, in our discussion section, links to another current theory of motor control, that of Optimal Feedback Control. We have included that paragraph here:

In this study, we concentrate on the neural responses to a visual shift in target location. However, there are many forms of perturbations to which subjects may need to respond during movements. For example, subjects also make corrective movements in response to tactile or proprioceptive information. If a change in target location is signaled via tactile information, subjects can respond with similar latency and accuracy as for visual target jumps (Pruszynski et al., 2016). Furthermore, subjects also make corrective movements to perturbations of the limb, typically at a latency even shorter than the latency for corrections in response to visual perturbations of the cursor or target (Cluff et al., 2014).

While the shortest-latency responses to mechanical perturbations likely reflect spinal circuitry (Ghez and Shinoda, 1978), long-latency mechanical perturbation responses can be quite sophisticated, taking task requirements into account (Scott, 2012). The Optimal Feedback Control paradigm suggests that a major function of preparatory activity is to set the feedback control policy for the upcoming movement (Scott, 2004; Todorov and Jordan, 2002). We found that movements are re-prepared following a target jump. One possible hypothesis would be that this change in task goal requires a similar change in control policy, therefore requiring re-preparation. A different form of perturbation, such as a small mechanical perturbation of the arm, may not require a change in policy and thus may not re-engage preparatory activity. Alternately, preparatory may need to be engaged for any change in the ongoing reach. Additional experiments under a wider variety of perturbations may help to shed light on this question.

The second major comment is the final argument and data analyses presented in relation to Figure 7. The goal is to separate the neural activity into two components, the neural activity that best separates the initial and final targets (C and E) and the remainder of the activity. Concerning Figure 7, it appears that several of the plots are either misplaced or potentially wrong (the reader assumes the former). Plot A is nearly identical to plot F and Plot B is nearly identical to Plot D in both shape and variability. Therefore, it seems that plot D and F are incorrect, that is plots for the two monkeys have been switched. (As stated above the other possibility is that the plots are incorrect, but the reader assumes just switched.) The second issue is that for both monkeys, the plots of the dimension separating the initial from final reach (C and E) is apparently only a small fraction of the overall mean neural distance (A and B). The questions are whether the fraction of the firing in this dimension is actually only a small component of the firing, can this be better quantified, and, if so, how can we be certain of its overall importance to the behavior.

Thank you for raising this important question regarding our analyses related to neural responses in the “separating” and “non-separating” dimensions. We have significantly modified our analyses in this section of the paper, in response to your and other reviewers suggestions.

In particular, instead of considering neural responses to the target in the “separating” and “non-separating” dimensions, we divided neural “where”-related activity into two categories of dimensions: the “preparatory dimensions”, which distinguish between different targets during the preparatory period, and the “movement” dimensions, which distinguish between targets during the movement epoch. Critically, these dimensions are orthogonal and can thus be analyzed separately. The activity in these neural dimensions explains the majority of neural cross-condition variance (see, for example, Figure 6), so we have more confidence of the importance of these dimensions for behavior.

Whereas our previous version of this paper only considered responses in the “separating” dimension (likely somewhat similar to a subset of our “movement” dimensions), we can now comment on the role of movement and preparatory signals. We can thus observe that motor cortex responds to a target jump in both movement and preparatory dimensions, but that it has a more “direct” response in the movement dimensions, transitioning relatively directly from one target’s pattern to the other. In comparison, the neural response in “preparatory” dimensions can be disproportionate; even if the distance between the first and second target’s response in

preparatory dimensions is not large (for example, for late target jumps), we nevertheless see a large response in preparatory dimensions. We hope that these additional analyses help to shed light on this process and give more clarity to the role of these signals beyond our previous “separating versus non-separating” dimensions analysis.

The third major comment concerns the analysis of the initial reach angle and analyses presented in Figure 2B. The question is how was the data normalized and is the normalization affecting the conclusions? It appears that the normalization was based on the degree of the jump (i.e. 180 vs 45 degrees). However, if this is the correct understanding, the normalization appears to be distorting the data. For example, while the plots for 180 degrees show few in-between values, the jumps for 45 degrees appear distributed with more intermediate values. However, normalizing by 180 will make the initial angles small, actually 4 times smaller than normalizing by 45 degrees. For example, a value of 0.1 in the 180 degree plot is equivalent to 0.4 in the 45 degree plot. Finally, the data in Figure 2 was stated to be fit to a “sinusoidal” function. Clearly, the shape of these plots are not sinusoidal. Were the data actually fit to a sigmoid?

You raise an important point regarding the normalization of the initial reach angles. As you intuited, the normalization was based on the degree of the target jump. We believe that two factors are at play in the results that you observe. First, as you point out, the “size” of the error will be magnified for the 45-degree jumps relative to the 180 degree jumps. In particular, this is likely why the cloud of points on the flat portions of the sigmoids is larger for the 45 degree than for the 180 degree case. Secondly, however, is the question of why there are some “intermediate” reach angles for the smaller jumps, but not for the 180 degree jump. This is actually a previously documented effect (see, e.g. Haith et al 2015), possibly having to do with whether or not an intermediate jump is a beneficial strategy during times of uncertainty.

To address this – to help compare the fits in a non-normalized fashion – we have added an additional supplemental figure (now Figure S2), which shows the same data as in Figure 2, but not normalized by reach angle. We maintain the original normalization for the main paper, as we feel that it helps to compare the time course of the behavioral response to the different jumps between angles.

Finally, you are correct that the fits were sigmoidal, not sinusoidal. We have corrected the misstatement in the text. Thank you.

The final general comment is that the manuscript will be a challenge for non-specialists. Particularly for Figure 5 and beyond in which the neural state concepts and dimensional analyses are introduced and used. These concepts and analyses rely heavily on the authors’ previous publications, which in themselves are challenging. An effort should be made to better explain the neural state and dimensional analysis as this would increase the readability of the manuscript.

Thank you for raising this concern; we truly attempt to always address and make things as broadly accessible as possible within word-limit restrictions. But we must do even better. We have significantly rewritten our Introduction, and we have included a conceptual figure to attempt to explain some of the concepts we explore more thoroughly for a broad audience. We include the figure here for your reference.

Cartoon of neural hypotheses. (A) To identify important signals in the neural population, we can project them into new dimensions. Activity in each dimension is calculated as a weighted average of the firing rates of all neurons. In motor cortex, there tend to be separate dimensions active during movement preparation and movement generation (which tend to have different firing rates across different reaching directions), as well as a “trigger dimension” which changes in a consistent manner prior to movement for all reach directions. (B) “Direct Response” hypothesis, early jumps: For jumps which occur while the motor cortex is still in the preparatory period (before or just after the go cue), neural activity after a jump (dotted black) should transition from preparing a reach to the first target (red) to preparing a reach to the new target (blue). Because the neural correction to the target jump is completed before movement onset, target-jump activity in movement dimensions should be similar to a reach to the new target. (C) “Always Prepare” hypothesis, early jumps: For jumps which occur while the motor cortex is still in the preparatory period (before or just after the go cue), the neural correction is in the preparatory dimensions, so the neural predictions of the direct response hypothesis and the always prepare hypothesis are the same. (D) “Direct Response” hypothesis, late jumps: For target jumps which occur close to the onset of movement, there is no difference in the preparatory space between the pattern of activity for the first (red) and second jumps (blue). Under the “direct response” hypothesis, we would therefore expect to see the response to the target jump occur exclusively in the movement-related neural dimensions. (E) “Always Prepare” hypothesis, late jumps: Under the “always prepare” hypothesis, motor preparation must be re-engaged following a target jump. We would thus expect to see a target jump response in the preparatory dimensions, even though these dimensions are not ordinarily active during movement.

Specific Comments:

1. The description of the times of the analyses in lines 515 to 518 is confusing and should be restated.

Thank you for letting us know where our methodological descriptions were unclear. In this current paper version, we are no longer including this analysis.

2. Why is there such a large difference in the number of datasets for the two monkeys?

Why did monkey K with the linear arrays only have 3 datasets? Does that mean just three recording days, which seems very limited. A clearer definition of a “dataset” would be helpful.

For Monkey K, we included all recording days we had from this task; unfortunately, shortly after we began this batch of experiments, his health decayed and he needed to be euthanized. However, we note that two “datasets” per monkey with chronic electrode arrays is a fairly typical number (e.g. Churchland et al 2012, Ames et al 2014), so Monkey K’s dataset numbers are in line with previous studies. In addition, because the arrays are not moved from day-to-day, the inclusion of additional days does not necessarily increase the pool of neurons included in the study. The same units may be recorded across multiple days.

For Monkey S, we instead used 16 channel U-probes, which were acutely lowered each day. Monkey S therefore required more days to achieve a similar number of total units recorded. The result is that the different monkeys have different tradeoffs in recording methods: Monkey S has fewer units per day, but more total days, whereas Monkey K has more units per day, but fewer days of recordings.

Thank you for pointing out our lack of clarity around this issue. We have included explicit text in our methods section to address our definition of “datasets” and how the monkeys’ data differs.

3. Several of the statistical analysis rely on sequential testing of time points to determine the timing of a change. However, sequential testing has an inherent problem with multiple testing on non-independent data, i.e. if you test enough times you will get a significant change. How was the multiple testing problem controlled for?

Thank you for raising this important point. Indeed, care must always be taken when performing statistical tests on multiple data points. We would like to note that here the questions addressed by these analyses were not “is there a time point at which the difference is significant” (which would become more and more likely as more times were added, without multiple comparisons controls), but rather “At which time point does the data change from being significant to not being significant, or vice versa?” In this case, adding more time points would not affect the results, as the question is not regarding significance per se, but rather the time at which significance changes.

We would also like to note that we have changed the primary focus on the paper from the timing of the neural responses to the pattern of those responses in different dimensions: the trigger dimension, the preparatory dimensions, and the movement dimensions. We have thus eliminated our analyses of the neural divergence/convergence after a target jump in order to make space for this change of focus. We therefore no longer use or rely on sequential applications of statistical tests.

4. Both primary motor cortex and dorsal premotor cortex neurons were recorded and analyzed together. Are there any differences in the neural activity in the two areas? Wouldn’t one expect PMd neurons to respond earlier and therefore, may respond differently to the target jump.

Thank you for this important question. We have added a supplementary figure (Figure S5) analyzing the contribution of PMd and M1 to the different dimensions (trigger, preparation, and movement) that we examined in this task. While both regions contributed to all dimensions, we found that they contributed similarly to trigger dimensions (with a slight preference for M1), while PMd contributed preferentially to preparatory dimensions, and M1 contributed preferentially to movement dimensions.

5. On page 11, line 229 the prediction median was 39% and 15% for the two monkeys. These numbers seem low, particularly so for the second animal. It suggests that the neural state analysis represents only a fraction of the information needed. This needs additional comments.

We have slightly how this analysis is performed, which improves our prediction abilities. First, we now attempt to predict the initial angle of the target jump trials, rather than the “distance reached to the wrong target.” This brings the analysis more in line with our behavioral analyses from Figure 3 (previously, Figure 2). Second, instead of asking the state of the jump dimension at the time of the target jump, we compare the timing of the target jump “zero” crossing (a measure of the onset of neural reach initiation) and the timing of the target jump. This enables us to more directly assess the relative timing of the target jump and neural trigger signal. After these modifications, our % variance explained is now 54% (25%) for monkey K (S). We include the text of our relevant results section here:

Across all jump conditions, the relative timing between the neural trigger and the target jump was able to predict a median of 52% (21%) of the variance in the initial reach direction for Monkey K (S), assessed using leave-one-out cross-validation of a sigmoid fit. Using the same fits, we also analyzed classification accuracy, to determine how well our fits could predict whether a reach would be toward the first or second target. This is a slightly easier problem, as classification is simply trying to determine whether the initial reach angle is closer to the first or new target, rather than the exact angle. For our classification problem we were able to predict the reach behavior with a median of 87% (74%) accuracy for Monkey K (S).

The major distinction between the predictive abilities of the two monkeys likely comes down to how many simultaneously recorded units we have from each animal. Because Monkey K had chronically-implanted arrays, we were able to record from many more units each day than for Monkey S. We have included a supplementary figure which compares our ability to isolate the trigger dimension (as assessed on held-out non-jump trials) with different numbers of units; this analysis suggests that performance scales similarly with the number of simultaneous units for each monkey.

6. In Figure 6F, there are negative R2 values? How is this possible?

We assess the performance of our fits using data which was not used to train the predictor: this means that the R2 values can be negative, indicating that the predictor did not generalize well to trials it did not see during training. Essentially, a negative value implies that our predictor does a worse job than simply predicting the mean. To make this more clear, we have changed our axis label in Figure 6 E-F to read "Generalization R^2 ," and expanded the description of our fits in the

methods. The section now reads: To assess the usefulness of the trigger dimension in predicting behavior in jump trials, we found, for each trial, the time that neural activity in the trigger dimension first crossed zero following the go cue. We compared this time to the time of the target jump, to see on each trial whether the target jump preceded or followed the neural trigger event. We then used this time offset between neural trigger and target jump to predict the initial angle of the reach, using a sinusoidal fit. Performance was assessed using leave-one-out cross-validation, in which we trained a sigmoid on all trials but one, and then predicted the behavior on that trial, repeating the process for each trial. R^2 values were calculated based on these left-out predictions. Note that because the predictions are calculated for trials that the classifiers don't have access to, the prediction can have a negative R^2 , indicating that the classifier performs worse than just guessing the mean.

7. Figure 5A shows the data for the argument that the neural response time does not differ between the onset and jump response, therefore, the sensory processing is not different in the two conditions. However, in both monkeys, the target jump response rises to a higher level and remains higher. The accumulation of neural activity over time is greater for the target jump. Isn't it possible that the detection or perception of the target depends on the accumulated response and, therefore, the accumulation

We have now broken down the neural response to the target jump into greater detail; instead of simply classifying the response as occurring in “separating” or “non-separating” dimensions, we examine the neural response in “preparatory” or “movement” dimensions. We find that the movement-dimension response tends to be relatively direct, while the preparatory response can often be larger than expected based on the non-jump trajectories. It is possible that this preparatory response is required in order to properly “detect” the target jump, though preparatory responses can involve both increases and decreases in neural activity.

Reviewer #4 (Remarks to the Author):

In this paper, Ames and colleagues investigate the difference in reaction time between trials where the target jumps just after the go cue and trials where it did not. It had been reported that reaction times are lower for jump trials than non-jump trials but the reason for this difference remains unclear. The authors found that the decoupling between the process of where to reach and when to reach was responsible for this effect and that the facilitation effect found in the jump trials was due to the fact that the go-process was already active and need not to be restarted later.

I like the approach and the study brings new insights to an interesting phenomena. Yet, I think that there are some conceptual problems with the study and that the analysis of the data could be better performed.

Thank you for this accurate and comprehensive summary.

Major concerns.

1) The authors compared the activity in jump and non-jump trials. Yet, the activity after the first target appearance must be confined to the null-space (Kauffman) in order to prevent any motion from the arm before the go cue. In contrast, after the target jump, the neural activity should not be confined to the null space and can already take place in all dimension given that the movement must not be prevented. Therefore, I wonder whether the neural activity after the presentation of the first target and the neural activity after the jump are really comparable. Indeed, the dimensions that can be explored in these two moments are different (null space vs. the whole space). The ideal control would have been non-jump trials without delays where neural activity can also evolve in the null and movement sub-spaces but these trials are absent from this report.

Thank you for raising this insightful and important question. Indeed, it is not a given that the response to a target jump will occur in the same neural dimensions as a delay-period response to target onset. As you note, during the delay the monkey is constrained to limit his neural response to muscle-null dimensions only, whereas during movement this constraint is lifted. We have substantially changed the focus of this paper to concentrate on exactly this question: to what degree is the neural response to the target jump performed by the same neural “dimensions” as during delayed-reaching. Because we performed a delayed-reaching task, we were able to use the different epochs of non-jump trials to find an orthogonal set of dimensions corresponding to neural preparation (e.g. delay period activity) and movement execution (e.g. movement-epoch activity). We then demonstrate that neural activity following a target jump indeed responds in each of these groups of dimensions. As you note, this need not have been the case, as the delay-period activity could be specific to delayed reaching rather than a general, required step for motor preparation. We hope that you will find these additional analyses compelling, and we thank you for your suggestion to examine this aspect of our data in greater detail.

2) The first model presented by the authors (faster processing along the visuomotor transformation) looks like a strawman model to me. Indeed, if we look at the effect of attention on sensory processing, people do not report faster activity (despite the fact that attention leads to

faster reaction time) but to higher firing rate at the single neuron level and more correlated firing rates at the population level (Reynolds, John H. and Leonardo Chelazzi. 2004. "Attentional Modulation of Visual Processing." *Annual Review of Neuroscience* 27(1):611–47; Maunsell, J. H. R. and E. P. Cook. 2002. "The Role of Attention in Visual Processing." *Philosophical Transactions of the Royal Society B: Biological Sciences* 357(1424):1063–72). Nobody has ever demonstrated that the faster reaction time due to attention were due to faster transmission time. Therefore, the first model proposed by the authors does not make sense to me. Interestingly, the authors observed one hallmark of attention in their data, an increased firing rate. I wonder whether the correlated activity also increases in their data after target jump as compared to after target presentation. If it were the case, the facilitation of reaction time after target jump would look a lot like an attentional effect at the neural level and this should be reported.

You raise an interesting point regarding our framing of our results regarding neural timing. Given your suggestions and those of the other reviewers, we have chosen to move away from an emphasis on the neural timing in this task, instead concentrating on the recruitment of different neural dimensions in response to the target jump. We believe that this change of focus should help to alleviate some of the concerns regarding potential interpretation issues in and around the response timing of neural activity in this task.

3) The analysis linked to Fig.6c and d seems very indirect to me. The authors want to look at (lines 208-210) "how close the monkey is to making a movement, and this readout should tell us (the same information) whether the monkey is reaching to the initial target or to the final target." Yet, they do not report correlation between those two variables but between other variables that are somehow related to them. Indeed, rather than looking at the time of the target jump with respect to the neural trigger moment, they look at the state of the trigger signal at the time of target jump. This does not seem to be a good variable as it looks very variable (Fig 6a and b). I don't see any reasons why the time relative to the time of neural trigger cannot be directly measured. The other variable was a measure of the reaching movement in the wrong direction. The authors chose for the travelled distance. Yet, this variable can be highly influenced by the vigor of the movement. Comparing the actual reaching direction to the reaching direction of non-jump trials to the final target is certainly a better measure to quantify how far from the final target the monkey is going towards.

Thank you for your important suggestions. We have changed the panels you have pointed out to reflect your suggestions. First, we made a slight alteration in the way we predicted behavior based on the trigger signal. Rather than looking at the state of the trigger signal at the time of the target jump, we instead identified the trigger signal's zero-crossing time for each trial, and compared that time to the time of the target jump. This more precisely measures the variable in question: did the target jump before or after the monkey began the neural process of initiating the first movement? Second, we predicted the initial angle of the reach rather than the distance reached to the wrong target. While these two variables are related, predicting the initial angle enables a closer correspondence with previous behavioral analyses (such as in Figure 3 (previously Figure 2)), and is less likely to be affected by additional variables like reach velocity, as you point out.

4) Detection of the different timing measure are based on the significance and non-significance of t-tests. This method is completely inappropriate and suffers from the fact than one outlier could make the test non-significant by increasing the SD. The authors should rather use the ROC analysis technique first used by Brian Corneil (Corneil, Olivier and Munoz, Neuron, 2004). This technique is also used by Steve Scott in his papers. In comparisons to what the authors have been using to detect differences between curve, the ROC technique will be much less sensitive to individual outliers and is therefore better suited to detect timing of different events. All the corresponding analyses need to be changed.

Thank you for your detailed suggestion on how to improve our statistical tests. Because we are no longer focusing on the timing of the neural responses, we are no longer including analyses which involve repeated statistical tests across time. Instead, where appropriate, we perform a single statistical test at time points of interest. For example, in our investigation of whether the response to the target jump was larger than expected in the preparatory dimensions, we used a Wilcoxon rank-sum test to compare the neural distance to the neural trajectory for reaches the “new” target between target-jump reaches and non-jump reaches toward the “original” target. We found that, for incorrectly-initiated target jumps, the response in preparatory dimensions during movement was much larger than would be predicted based on the distance between the non-jump trajectories (Figures 7-8). Because this statistical test is 1) non-parametric and 2) applied only at one particular time, we believe that this will alleviate the important concerns related to our investigation of the timing of neural responses in our previous paper version.

5) Analysis related to Fig 7 c-f is really complicated to follow because the criteria for convergence for the different panels are different . In 7c and e, 1 means neural activity close to initial target and -1 means close to final target (but this isn't explained properly in the main text). In contrast for 7d and f (and also 7a and b), convergence corresponds to going to zero. I feel that this difference on the definition of convergence in the same figure is really confusing. Moreover, the description and meaning of these curves need to be improved (e.g. by adding for 7c and e that 1 corresponds to neural activity for movements towards the initial target and -1 towards final target). Here are several questions related to the above analysis (Fig.7):

1) I wonder whether the faster detection of changes in neural activity in the separating dimension could be due to the higher signal to noise ratio there. Clearly, with the t-tests used here, a better SNR would yield an earlier detection of the convergence. So, is the SNR comparable in the two dimensions?

2) The criteria for detecting convergence is very different in the two dimensions. For the separating dimension, crossing zero is sufficient (even though the convergence is not completed yet). For the other one, the minimum distance should be zero. I believe that the authors could use the same criteria in both cases (minimum distance on each time point could be computed in the separating dimension as it was done in the null dimension). This would yield a fairer comparison for the timing of convergence between the two dimensions.

The points that you raise above are quite insightful, thank you for bringing them to our attention. We agree that the manner in which we calculated convergence in the previous version of the paper was potentially problematic, as it was performed in different ways in different spaces, and involved the comparison of activity in a single dimension to activity in a high-dimensional space. In our new paper variant, we split neural activity into preparatory and movement dimensions,

(rather than the “separating” dimension and “all other” dimensions), and analyze neural responses in these dimensions in the same way. In each case, we look at the neural distance between target jump conditions and the non-jump reaches to the “correct” target as a function of time. The comparison in each case is between the neural distance in the jump conditions and the neural distance for the different non-jump conditions. Because these trajectories are analyzed in the same spaces and in the same manner, we believe that this helps to address the methodological challenges you raise here.

3) The authors should comment on the relative timing of neural activity convergence and the timing of movement onset. Right now, it is difficult to know whether this convergence happens before or after movement onset as everything is aligned on target jump time.

Thank you for this suggestion. In our figures, we now align all conditions in the same way; to target onset, go cue, and movement onset. This should enable readers to more easily examine the differences between target jump conditions and non-jump conditions as a function of time in the trial.

Reviewers' Comments:

Reviewer #1:

Remarks to the Author:

Reviewer #1,

Following my previous comments on the paper's previous version and the improvements carried out in the current version, I have the satisfaction to admit that the paper meets now all my requirements. The hypotheses in the paper are now well balanced and the additional figures improve its didactic value. The paper leverages the advantages of using neural dimension analysis to study online-reaching corrections and identifies motor preparation and execution as independent but parallel processes.

Reviewer #2:

Remarks to the Author:

The submitted paper by Ames and colleagues is a substantially revised version of a manuscript I previously reviewed. The new version of the paper is much improved, much sharper. This is a timely paper that very nicely leverages modern analytical techniques to provide new insight into the neural control of movement, specifically the rapid updating of reaches that occur in the context of goal target jumps. I have several comments related to the assumptions of the approach and point out the need for sharpening the presentation in terms of fit with the previous literature.

Major Things:

1. It seems to me that the authors are making one very important assumption. That is, they extract preparatory dimensions and movement dimensions from the baseline data and then assume that these hold in the context of target jumps. Now, I am not saying this is not true and perhaps I am wrong but I don't see an obvious way that this can be substantiated. Let me emphasize that I think the authors are making a reasonable assumption and that this is not a deal breaker. However, I think that, at the very least, the authors need to comment on this possibility as a caveat with respect to their experimental design (introduction) and interpretation (discussion).

2. A key weakness of the present Discussion is the lack of synthesis in the context of which parts of the nervous system are actually contributing to the phenomenon under study. This is glaring because of the nature of the citations being used and because the general consensus in the field would be to consider the role of posterior parietal cortex and brainstem in this respect. All the authors have right now is some throwaway statement (Line 424-427). This is not sufficient because alternative data exists which places this work in a very specific context. I describe a bit below.

2a. If you look at the work of Goodale, Perlisson, Preblanc, (as cited and follow up work) they make a big deal about the responses being subconscious. They also have work with patients that have posterior parietal lesions showing specific dysfunction in these tasks. How does this fit with the M1-centric view being put forth here?

2b. You have not considered the role of the brainstem and, in this domain, you have missed the work of Brian Day. He has over many years shown some beautiful work in the target jump domain (e.g. Day and Lyon, EBR, 2000). What's interesting about his work is that he suggests that rapid updates to target jumps are subcortical. For example, he shows that the target jump effect is not sensitive to the number of possible target jump locations (same as Pruszynski et al., Current Biology,

2014, which you did cite). Moreover, he has a wonderful paper with an acallosal person showing a pattern of results consistent with a brainstem pathway for target jumps. (Day and Brown, Brain, 2001). There is additional relevant work that points to the brainstem that should be considered as well. For example, Pruszynski has a paper in European Journal of Neuroscience (2010) that shows that the earliest response to a target jump is time locked to the new target appearance at a latency of < 100ms. If this is the fastest set of muscle activity driving arm responses to target jumps, it seems like cortex may not play much of a role. He also suggest brainstem specifically that this is a tecto-reticulo-spinal response and, indeed, Klaus Peter Hoffman and others have shown projections for superior colliculus to proximal arm muscles.

3. Related to the above, I am a bit confused by the chose citations. I mean, there are many studies looking at target jumps and its very unclear to me why the particular ones are being chosen. Two specific points here. Early in the introduction the notion of target jumps is introduced via citations to Georgeopoulos, Soething, Archambault (Line 68), later the authors make explicit mention of previous papers using this specific paradigm which then cites a dozen papers (Line 79-80) and then they switch when talking about target jumps again to the likes of Goodale, Pelisson, Preblanc (Line 144). Then in there is also the work of Haith thrown in occasionally and then yet another subset of target jump work in the Discussion (408-9). Its all a bit confusing frankly because not all target jump approaches are the same and its unclear why some are cited in one place and not the others. I would suggest to the authors that they stick to those papers upon which their paradigm is built in most instances and then in the Discussion bring in these alternative approaches being very clear how they are different and what the present work has to say about them (e.g. as they come up in the context of my comments 2a,b.

Other Things:

Abstract. Online is a bit jargon, I would spell it out for the person at this point in the paper.

Line 64. Remove the However.

Line 66. Franklin and Wolpert is "visual feedback of hand position".

Line 157. Please provide the statistic, degrees of freedom in addition to the p-values. (Here and everywhere statistical tests are performed).

Line 211: This is one place where the assumption comes in (see #1 above).

Line 231: I don't understand why zero is meaningful value here. Please elaborate.

Line 343: remove "is"

Figure 5E,F. for these histograms it seems there are too few bins.. approx sqrt(n). Anyway, it looks a bit strange.

Figure 7. So cool!

Reviewer #3:

Remarks to the Author:

General Comments

The authors have been very responsive to the concerns and questions raised by both this and the other three reviewers. As a result, the manuscript presentation has improved as have the presentation of the results and analyses. The analytical and statistical approach is very strong. The addition of Fig. 1 that graphically outlines the underlying concepts are very helpful visual to introduce a complex topic and analyses. Overall, the manuscript is very much improved and makes an important contribution to how we understand neural preparation for movements.

This reviewer has a few remaining questions. The first concerns the differences in the reaction time in a normal reach versus a re-directed reach. The former is approximately 330 msec. and the later 160 msec. This is an important concept in the manuscript, but the extra 170 msec. that it takes to generate a normal reach is never quite addressed. It would be of value for the authors to articulate their view on this.

Specific Comments

1) In several of the figures, the lines used are quite thin (for example in Figs. 6, 7, and 8). Thin dotted lines are difficult to see as are differentiating the colors of thin lines. Suggest making the lines thicker/bolder where possible.

2) It is stated that the dimensions differ on each day (lines 275-277). How different are they from day-to-day? It would seem that in a highly trained animal, similar neurons would be used in the reach and the dimensions would not vary greatly. Is the variability because a different set of neurons are recorded each day? What are the implications for this variability?

3) The average neural activity in the trigger direction is much less prominent in monkey S versus K (Fig. 5A vs B). The response is actually quite small in monkey S. What are the implications of this smaller neural profile in monkey S? Does this have a behavioral consequence?

Reviewer #4:

Remarks to the Author:

This is the second version of this paper by Ames et al. The paper is much changed with respect to the first paper and is much improved. I still have a few comments though:

Major comment

1. In Fig.1, the authors depict that, following the "always prepare" hypothesis, the extra-preparation takes place at the same time as the activity in the movement dimensions. Is this true or does this extra-preparation takes place before activity in the movement dimensions. There are some argument in Haith's work (Haith, Adrian M., Jina Pakpoor, and John W. Krakauer. "Independence of movement preparation and movement initiation." *Journal of Neuroscience* 36.10 (2016): 3007-3015.) that movement preparation needs to be done before movement begins (this is also mentioned in the paper) while other works suggest that movement preparation can be finished during movement execution, hence reducing reaction time (Orban de Xivry, Jean-Jacques, Valery Legrain, and Philippe Lefevre. "Overlap of movement planning and movement execution reduces reaction time." *Journal of neurophysiology* 117.1 (2016): 117-122.). The latter is in accordance with the ballpark idea of Green (Greene, Peter H. "Problems of organization of motor systems." *Progress in theoretical biology* 2 (1972): 303-338) where the planning stage can bring the system in the right states before the feedback control policy can easily correct for any deviation from the goal. I would like the authors to

comment in their manuscript whether they think that activity in the preparation and movement dimensions occur in parallel or sequentially. Ideally, this would be supplemented by some additional analyses. It is currently unclear whether the activity in the movement dimensions is delayed in jumped trials compared to non-jumped trials or not. Is there a correlation between the extra time needed to prepare the movement and the start of the activity along the movement dimensions? I believe that this is an important topic that would increase the impact of this paper.

Minor comment

1. I believe that it would be clearer to replace the terminology of correct/incorrect trials by towards final/towards initial target trials. This would make the graphs much easier to understand given that the authors refer to the distance to final target trajectory. In this case target 1 and 2 are referred to as initial and final targets. It is currently difficult to understand the link between correct/incorrect trials, initial/final targets and target 1/target 2. That's a lot of different terminology for two targets...

Color guide:

Black = reviewer comment

Blue = response to reviewers

Green = excerpt from paper

Reviewers' comments:

Reviewer #1 (Remarks to the Author):

Reviewer #1,

Following my previous comments on the paper's previous version and the improvements carried out in the current version, I have the satisfaction to admit that the paper meets now all my requirements. The hypotheses in the paper are now well balanced and the additional figures improve its didactic value. The paper leverages the advantages of using neural dimension analysis to study online-reaching corrections and identifies motor preparation and execution as independent but parallel processes.

Thank you. We appreciate your assistance in revising and improving this manuscript.

Reviewer #2 (Remarks to the Author):

The submitted paper by Ames and colleagues is a substantially revised version of a manuscript I previously reviewed. The new version of the paper is much improved, much sharper. This is a timely paper that very nicely leverages modern analytical techniques to provide new insight into the neural control of movement, specifically the rapid updating of reaches that occur in the context of goal target jumps. I have several comments related to the assumptions of the approach and point out the need for sharpening the presentation in terms of fit with the previous literature.

Major Things:

1. It seems to me that the authors are making one very important assumption. That is, they extract preparatory dimensions and movement dimensions from the baseline data and then assume that these hold in the context of target jumps. Now, I am not saying this is not true and perhaps I am wrong but I don't see an obvious way that this can be substantiated. Let me emphasize that I think the authors are making a reasonable assumption and that this is not a deal breaker. However, I think that, at the very least, the authors need to comment on this possibility as a caveat with respect to their experimental design (introduction) and interpretation (discussion).

You raise an important point, which deserves to be highlighted better within the text. As a result, we have added the following to the introduction:

Furthermore, defining what “re-preparation” means in the context of online reach correction is itself a challenge. Based on the finding that preparatory and movement activity occur in distinct neural subspaces (Elsayed et al., 2016), we operationally define motor preparation to be neural activity in dimensions that are normally active during the delay period. If, following a target jump, movements are re-prepared in the same way as during normal reach preparation, we would expect to see a re-entry into these dimensions. If, on the other hand, the new movement is either not prepared or is re-prepared in a fundamentally different manner, we would expect to see no special activity in these putatively preparatory dimensions.

To the discussion, we added a new section, entitled “The form of preparatory activity”:

In this study, we defined preparatory dimensions as the dimensions which are selectively active during the delay period. This definition encompasses two assumptions. First, it assumes that neural activity during the delay period serves a preparatory role. The preparatory role of delay period activity has been heavily studied. Having a delay period speeds reaction time (Rosenbaum, 1980; Riehle and Requin, 1989; Crammond and Kalaska, 2000), and trial-to-trial variations in delay period activity are related to variations in reaction time (Churchland et al., 2006a; Afshar et al., 2011). Our finding that these dimensions are also recruited following a target jump adds further evidence that their role is important for programming and updating movement commands.

Our second assumption regarding motor preparation is that preparation will always involve the same dimensions. This again seems reasonable given prior studies of these dimensions. For example, the same preparatory events are activated prior to movement not only during delayed reaching, but also under several additional behavioral paradigms, including a self-timed movement task and a quasi-automatic response task (Lara et al., 2018). However, the same preparatory dimensions need not necessarily have been involved in a response to a target jump. For example, if we had found that neural activity did not re-enter these preparatory dimensions following a target jump, it

could have indicated either that re-preparation does not occur or that re-preparation leverages a different set of neural dimensions. Because we did observe a neural response in these putatively-preparatory dimensions, this supports the hypothesis that these dimensions are, in fact, used not only during the delay period but also during last-moment re-preparation of movements and online reach correction.

2. A key weakness of the present Discussion is the lack of synthesis in the context of which parts of the nervous system are actually contributing to the phenomenon under study. This is glaring because of the nature of the citations being used and because the general consensus in the field would be to consider the role of posterior parietal cortex and brainstem in this respect. All the authors have right now is some throwaway statement (Line 424-427). This is not sufficient because alternative data exists which places this work in a very specific context. I describe a bit below.

2a. If you look at the work of Goodale, Perlisson, Preblanc, (as cited and follow up work) they make a big deal about the responses being subconscious. They also have work with patients that have posterior partial lesions showing specific dysfunction in these tasks. How does this fit with the M1-centric view being put forth here?

2b. You have not considered the role of the brainstem and, in this domain, you have missed the work of Brian Day. He has over many years shown some beautiful work in the target jump domain (e.g. Day and Lyon, EBR, 2000). What's interesting about his work is that he suggests that rapid updates to target jumps are subcortical. For example, he shows that the target jump effect is not sensitive to the number of possible target jump locations (same as Pruszynski et al., Current Biology, 2014, which you did cite). Moreover, he has a wonderful paper with an acallosal person showing a pattern of results consistent with a brainstem pathway for target jumps. (Day and Brown, Brain, 2001). There is additional relevant work that points to the brainstem that should be considered as well. For example, Pruszynski has a paper in European Journal of Neuroscience (2010) that shows that the earliest response to a target jump is time locked to the new target appearance at a latency of < 100ms. If this is the fastest set of muscle activity driving arm responses to target jumps, it seems like cortex may not play much of a role. He also suggest brainstem specifically that this is a tecto-reticulo-spinal response and, indeed, Klaus Peter Hoffman and others have shown projections for superior colliculus to proximal arm muscles.

3. Related to the above, I am a bit confused by the chosen citations. I mean, there are many studies looking at target jumps and its very unclear to me why the particular ones are being chosen. Two specific points here. Early in the introduction the notion of target jumps is introduced via citations to Georgeopoulos, Soeething, Archambault (Line 68), later the authors make explicit mention of previous papers using this specific paradigm which then cites a dozen papers (Line 79-80) and then they switch when talking about target jumps again to the likes of Goodale, Pelisson, Preblanc (Line 144). Then in there is also the work of Haith thrown in occasionally and then yet another subset of target jump work in the Discussion (408-9). Its all a bit confusing frankly because not all target jump approaches are the same and it's unclear why some are cited in one place and not the others. I would suggest to the authors that they stick to those papers upon which their paradigm is built in most instances and then in the Discussion bring in these alternative approaches being very clear how they are different and what the present work has to say about them (e.g. as they come up in the context of my comments 2a,b.

Thank you for these important points (2a-b,3). We have made several changes to our discussion of the relevant literature in order to better highlight the current state of the field, and

to place our work in context. First, we streamlined our citations in the introduction and throughout the paper, in order to concentrate on the most-relevant studies to set up our experiment. In the discussion, we added a section called “Implications for the mechanism of target jump response,” in which we discuss several models of how the brain performs online reach corrections, including the studies which you helpfully pointed out. We then place our findings in the context of these prior results and suggest how the cortical and subcortical mechanisms of online reach correction might be reconciled. We include this section of the discussion below:

A major behavioral signature of target jump responses is that the reaction time to the jump is faster than the normal reaction time for point-to-point reaching. This has been observed across a wide variety of behavioral paradigms (Carlton, 1981; Soechting and Lacquaniti, 1983; Prablanc and Martin, 1992; Day and Lyon, 2000; Pisella et al., 2000; Pruszynski et al., 2010; Archambault et al., 2011; Saberi-Moghadam et al., 2016), and is similar in magnitude to the RT decrease observed in metronome tasks (Ghez et al., 1997; Haith et al., 2015). Two major mechanisms have been proposed to explain this RT speedup.

The first mechanism is that subcortical areas may play a special role during online reach correction. This model is supported by several lines of evidence. First, reach corrections are quite fast (muscle responses can be found within 100 ms: (Pruszynski et al., 2010)), and can be performed subconsciously, for small target jumps that occur during the saccade (Pelisson et al., 1986; Prablanc et al., 1986; Prablanc and Martin, 1992). Second, initial target jump responses are the same regardless of subjects’ intention, and only later do task instructions (such as to “reach away” from the jump location) affect behavior (Day and Lyon, 2000; Johnson et al., 2002). Finally, a subject with callosal agenesis experienced an RT deficit when reaching to targets in the opposite hemifield during point-to-point reaches, suggesting that reach initiation normally involves cortical communication pathways. However, target jump responses were unaffected, suggesting a subcortical route for online reach correction (Day and Brown, 2001).

A second proposed mechanism is that motor preparation and execution, normally performed in sequence, instead overlap in time during online reach correction. Evidence that motor preparation and execution can be performed simultaneously come from a few sources. First, Haith and colleagues found that preparation time and initiation time were statistically distinct; indeed, subjects couldn’t link the two even when it would be beneficial to do so (Haith et al., 2016). Errors in initial reach direction occurred when the reach began before preparation was complete, suggesting that in these error trials, preparation and execution overlapped. Second, neural variability in PMd was reduced following a target jump, suggesting that this area, historically associated with motor planning, may be playing a role in speeding responses to target jumps (Saberi-Moghadam et al., 2016). Third, a study of reaching where accuracy constraints are only taken into account mid-reach also have a low RT (de Xivry et al., 2016). This suggests that reaches can be initiated with incomplete preparation, and prepared online under certain task setups. It is also worth noting that perturbations of the posterior parietal cortex interfere with the fast response to target jumps, suggesting that cortical visuomotor pathways are also involved in online reach corrections (Desmurget et al., 1999; Pisella et al., 2000; Gréa et al., 2002), as opposed to reach correction being exclusively the provenance of subcortical structures.

Our results directly examined preparatory and execution related signals in motor cortex, and we found that preparatory signals are re-engaged following a last-moment target jump. This provides new evidence that the same motor preparatory process plays a role not only in specifying a reach ahead of time, but also in online reach modifications.

Our results therefore support the model in which preparation and execution can be performed simultaneously during online reach correction. We observe no change in the role of motor cortex for online reach correction versus standard point-to-point reaching; instead, we see the same set of signals in motor cortex during both tasks. Why, then, is the reaction time so much lower for responses to a target jump than for responses to initial target appearance? Our data supports the hypothesis that “normal” reaction times principally measure the time to trigger the movement, whereas the target jump reaction time instead measures the time to prepare a movement. This preparation time can be quite fast, and thus is normally complete prior to triggering movement (Wong et al., 2015; Haith et al., 2016).

Our results are not necessarily incompatible with a role for subcortical pathways, however. One intriguing possibility is that a subcortical relay of visual information to motor cortex is also used during normal reaching. Our previous study found that target-related information reaches motor cortex approximately 50 ms after appearing on the screen, suggesting that a very fast pathway transmits these earliest signals. In contrast, the motor cortical response to the go cue is much slower, requiring at minimum 100 ms during a standard reaching task (Ames et al., 2014). If the decision of when to reach is more cortically-dependent, then this could explain results like the agenesis study described above (Day and Brown, 2001). For the subject with callosal agenesis, point-to-point reaching reaction times could be impaired when reaching to the opposite hemifield due to an increase in time to trigger the reach, instead of an increase in time to prepare the reach.

Other Things:

Abstract. Online is a bit jargon, I would spell it out for the person at this point in the paper.

Thank you for pointing this out. We have now modified “online reach correction” in the abstract to be “Mid-reach modifications.”

Line 64. Remove the However.

The sentence now reads:

Because these dimensions have primarily been identified and studied during delayed reaching, their role in a wider variety of behaviors remains unclear.

Line 66. Franklin and Wolpert is “visual feedback of hand position”.

Corrected.

Line 157. Please provide the statistic, degrees of freedom in addition to the p-values. (Here and everywhere statistical tests are performed).

We have added the degrees of freedom information to all of our parametric statistics (such as ANOVA).

Line 211: This is one place where the assumption comes in (see #1 above).

Thank you. We hope that our added text in the intro and discussion will help the readers keep this important issue in mind as it becomes relevant throughout the results.

Line 231: I don’t understand why zero is a meaningful value here. Please elaborate.

This change was made at the request of another reviewer. For the previous version of this analysis, we had examined the relationship between the state of neural activity in the trigger dimension at the time of the target jump and the subsequent reach angle, with very similar

results. This change brought the focus to the relationship between the timing of the trigger event and the timing of the target jump: the zero crossing reflects when the decoder changes from predicting that neural activity is not close to initiating movement to when neural activity is close to initiating movement.

To help explain these ideas to the reader, we have added the following sentence to that paragraph:

Note that the location of zero is decoder dependent, and reflects when the decoder decides that the trial is “close to initiating movement.”

Line 343: remove “is”
Corrected.

Figure 5E,F. for these histograms it seems there are too few bins.. approx \sqrt{n} . Anyway, it looks a bit strange.

Thank you for pointing this out. Because the monkeys have different numbers of conditions, we had set the bin size to be the same for both. However, we have now changed the bin size to yield \sqrt{n} bins across the range of values. We also now explicitly note the sample size for each monkey in the figure text.

R^2 values show generalization accuracy across $n = 20$ conditions (Monkey K) and $n = 68$ conditions (Monkey S).

Figure 7. So cool!
Thank you!

Reviewer #3 (Remarks to the Author):

General Comments

The authors have been very responsive to the concerns and questions raised by both this and the other three reviewers. As a result, the manuscript presentation has improved as have the presentation of the results and analyses. The analytical and statistical approach is very strong. The addition of Fig. 1 that graphically outlines the underlying concepts are very helpful visual to introduce a complex topic and analyses. Overall, the manuscript is very much improved and makes an important contribution to how we understand neural preparation for movements.

Thank you.

This reviewer has a few remaining questions. The first concerns the differences in the reaction time in a normal reach versus a re-directed reach. The former is approximately 330 msec. and the latter 160 msec. This is an important concept in the manuscript, but the extra 170 msec. that it takes to generate a normal reach is never quite addressed. It would be of value for the authors to articulate their view on this.

Thank you for this suggestion – we would indeed like to be clear about what our results say about this important behavioral effect. We have added a section to our discussion section, entitled “Implications for the mechanism of target jump response,” in which we place our results in the context of prior online reach correction studies. In particular, we tried to clearly state our understanding of why the reaction time is faster to a target jump than to initial target appearance:

Our results directly examined preparatory and execution related signals in motor cortex, and we found that preparatory signals are re-engaged following a last-moment target jump. This provides new evidence that the same motor preparatory process plays a role not only in specifying a reach ahead of time, but also in online reach modifications. Our results therefore support the model in which preparation and execution can be performed simultaneously during online reach correction. We observe no change in the role of motor cortex for online reach correction versus standard point-to-point reaching; instead, we see the same set of signals in motor cortex during both tasks. Why, then, is the reaction time so much lower for responses to a target jump than for responses to initial target appearance? Our data supports the hypothesis that “normal” reaction times principally measure the time to trigger the movement, whereas the target jump reaction time instead measures the time to prepare a movement. This preparation time can be quite fast, and thus is normally complete prior to triggering movement (Wong et al., 2015; Haith et al., 2016).

Specific Comments

1) In several of the figures, the lines used are quite thin (for example in Figs. 6, 7, and 8). Thin dotted lines are difficult to see as are differentiating the colors of thin lines. Suggest making the lines thicker/bolder where possible.

Thank you for this suggestion – we have made lines thicker in Figures 6, 7, and 8.

2) It is stated that the dimensions differ on each day (lines 275-277). How different are they from day-to-day? It would seem that in a highly trained animal, similar neurons would be used in the reach and the dimensions would not vary greatly. Is the variability because a different set of neurons are recorded each day? What are the implications for this variability?

As you intuit, the difference in dimensionality is related to the fact that a different set of neurons is recorded each day. It is likely that in a highly-trained animal such as we are using here, the same neural dimensions will be leveraged from day to day. However, our ability to sample those dimensions is limited to the units that we happen to record from. If we happen to record from several units with correlated activity, then we may only be sampling from a subset of the true underlying neural dimensions. This effect is reduced the more units we record from, as our likelihood of recording units with significant weights for each dimension is increased.

3) The average neural activity in the trigger direction is much less prominent in monkey S versus K (Fig. 5A vs B). The response is actually quite small in monkey S. What are the implications of this smaller neural profile in monkey S? Does this have a behavioral consequence?

This difference has more to do with the fact that more neurons are recorded on each day in Monkey K than in Monkey S. This means that neural dimensions will, on average, tend to be larger, as they involve the summed activity of more units. We have repeated our “subsampling” analysis for Monkey K to examine the effect of number of units on the size of the trigger dimension – picking random subsets of units from Monkey K and examining the size of the trigger signal that we find in these subsets. This is shown in the Figure below (which is now subplot C-D of supplemental figure S4.) We calculated the trigger signal as the average neural activity in the trigger dimension across all non-jump trials, calculated at each time point from 500 ms before movement onset to 500 ms after movement onset (as in Figure 5A-B). The size of this signal was defined to be the range of firing rates this average trace achieved. We find that the size of the signal is similar between monkeys when the number of units is held constant, suggesting that the difference in size is likely due to differences in number of units recorded per day, rather than a fundamental difference between signals for the two monkeys.

Left panel: each line is the average size of the trigger signal found from each dataset, for a given subsampled pool size. **Right panel:** The median size of the trigger signal, across all samples and datasets. Shaded regions show 10th and 90th percentiles of the distribution.

Reviewer #4 (Remarks to the Author):

This is the second version of this paper by Ames et al. The paper is much changed with respect to the first paper and is much improved. I still have a few comments though:

Major comment

1. In Fig.1, the authors depict that, following the “always prepare” hypothesis, the extra-preparation takes place at the same time as the activity in the movement dimensions. Is this true or does this extra-preparation take place before activity in the movement dimensions. There are some argument in Haith’s work (Haith, Adrian M., Jina Pakpoor, and John W. Krakauer. "Independence of movement preparation and movement initiation." *Journal of Neuroscience* 36.10 (2016): 3007-3015.) that movement preparation needs to be done before movement begins (this is also mentioned in the paper) while other works suggest that movement preparation can be finished during movement execution, hence reducing reaction time (Orban de Xivry, Jean-Jacques, Valery Legrain, and Philippe Lefevre. "Overlap of movement planning and movement execution reduces reaction time." *Journal of neurophysiology* 117.1 (2016): 117-122.). The latter is in accordance with the ballpark idea of Green (Greene, Peter H. "Problems of organization of motor systems." *Progress in theoretical biology* 2 (1972): 303-338) where the planning stage can bring the system in the right states before the feedback control policy can easily correct for any deviation from the goal. I would like the authors to comment in their manuscript whether they think that activity in the preparation and movement dimensions occur in parallel or sequentially. Ideally, this would be supplemented by some additional analyses. It is currently unclear whether the activity in the movement dimensions is delayed in jumped trials compared to non-jumped trials or not. Is there a correlation between the extra time needed to prepare the movement and the start of the activity along the movement dimensions? I believe that this is an important topic that would increase the impact of this paper.

Thank you for this important question. We agree that this is a key finding of our paper, and of potential great importance. Indeed, we do think that activity in preparation and movement dimensions are occurring in parallel during online reach corrections. The critical comparison can be most easily seen in Figure 8 : for late target jumps, at the time of movement onset, we see high neural distances between jump and non-jump conditions both in the preparatory space and the movement space (Figure 8, dashed lines). If the re-preparation occurred prior to activating the movement dimensions instead, then the distance should remain low in the movement dimensions until preparation is complete. We added the following sentences to the results section discussing this figure to point out this important comparison:

Furthermore, for late target jumps, note that at the time of movement onset, neural distance is not only high in the preparatory space, but also in the movement space (Figure 8 C-D, dashed line). This indicates that not only are preparatory dimensions re-engaged following a target jump, but also that preparatory dimensions and movement dimensions can be engaged at the same time. In other words, the monkey is simultaneously generating movement and preparing movement.

We have also included text in our discussion about the simultaneous occupancy of preparation and movement dimensions. While this is noted a few times in the discussion, we would like to especially point out its placement in our final summary, which begins, “To our knowledge, this is the first study to provide neural evidence of simultaneous motor preparation and movement execution.” Our concluding paragraph reads:

Beyond experimental convenience, however, our results suggest an important computational advantage to leveraging different dimensions for different computations: the brain gains the ability to perform computations serially or in parallel, depending on

which is more suited to the desired behavior. Increasing numbers of brain regions have been shown to leverage different dimensions for different computations (e.g. prefrontal cortex: (Durstewitz et al., 2010; Machens et al., 2010; Mante et al., 2013); posterior parietal cortex: (Raposo et al., 2014); motor cortex: (Kaufman et al., 2014, 2016; Elsayed et al., 2016; Gallego et al., 2017; Lara et al., 2018); fly antennal lobe:(Stopfer et al., 2003)). Separate computations being mixed at the level of individual units but separable at the level of the neural population thus appears to be a common feature of neural processing. Our results suggest that this separation of signals by dimensions may allow brain regions to alter the temporal relationships between computations: movement preparation and generation can be performed one after another, or at the same time. By studying additional brain regions under a variety of task requirements, the ability to alter the temporal relationships between computations performed in different dimensions may well be revealed as a unifying feature of neural processing.

Finally, we also slightly modified our abstract to bring these ideas into focus at the outset of the paper. The concluding sentences of our abstract now read:

After a target jump, neural activity responded in both preparatory and movement related dimensions, even though error in preparatory dimensions could be small at that time.

This suggests that the same preparatory process used in delayed reaching is also involved in reach correction. Furthermore, it indicates that motor preparation and execution can be performed simultaneously.

Minor comment

1. I believe that it would be clearer to replace the terminology of correct/incorrect trials by towards final/towards initial target trials. This would make the graphs much easier to understand given that the authors refer to the distance to final target trajectory. In this case target 1 and 2 are referred to as initial and final targets. It is currently difficult to understand the link between correct/incorrect trials, initial/final targets and target 1/target 2. That's a lot of different terminology for two targets....

Thank you for this suggestion. We have gone through the paper and standardized our terminology. We now refer to targets always as the "first" target or the "final" target, and correct/incorrect jump responses are referred to as "jump trials initiated toward the first target" and "jump trials initiated toward the final target"

Reviewers' Comments:

Reviewer #2:

Remarks to the Author:

The authors have addressed my concerns. Looks great.

Andrew Pruszynski

Reviewer #3:

Remarks to the Author:

The authors have responded thoroughly to my second level review of this manuscript and satisfactorily addressed my remaining concerns. In my opinion, the work is a novel and makes an important contribution to the motor control literature on how the motor cortex functions during both preparation for reach and online connections. The results and analyses are compelling. The conclusion that the same processes are used in both types of movements provides new insights into motor control process. The manuscript leads to the intriguing concept that motor preparation and execution are performed simultaneously in the motor cortex.

I have only one minor comment, neither the title nor the abstract state that the recordings were done in PMd and M1. This seems like essential information that should be included in one or the other.

Timothy J. Ebner

Reviewer #4:

Remarks to the Author:

I don't have any further comments

****REVIEWERS' COMMENTS:**

Reviewer #2 (Remarks to the Author):

The authors have addressed my concerns. Looks great.

Andrew Pruszynski

Thank you.

Reviewer #3 (Remarks to the Author):

The authors have responded thoroughly to my second level review of this manuscript and satisfactorily addressed my remaining concerns. In my opinion, the work is a novel and makes an important contribution to the motor control literature on how the motor cortex functions during both preparation for reach and online connections. The results and analyses are compelling. The conclusion that the same processes are used in both types of movements provides new insights into motor control process. The manuscript leads to the intriguing concept that motor preparation and execution are performed simultaneously in the motor cortex.

I have only one minor comment, neither the title nor the abstract state that the recordings were done in PMd and M1. This seems like essential information that should be included in one or the other.

Timothy J. Ebner

Thank you. We now include mention of PMd and M1 in the abstract. Line 34-35: In motor cortex and dorsal premotor cortex, we find that the neural activity that signals when to reach predicts monkeys' jump responses on a trial-by-trial basis.

Reviewer #4 (Remarks to the Author):

I don't have any further comments

Thank you.